# Greatwall-phosphorylated Endosulfine is both an inhibitor and a substrate of PP2A-B55 heterotrimers

**Byron C Williams[1], Joshua J Filter[1], Kristina A Blake-Hodek[1†], Brian E Wadzinski[2], Nicholas J Fuda[1], David Shalloway[1], Michael L Goldberg[1]\***

[1]Department of Molecular Biology and Genetics, Cornell University, Ithaca, United States; [2]Department of Pharmacology, Vanderbilt University Medical Center, Nashville, United States

**Abstract** During M phase, Endosulfine (Endos) family proteins are phosphorylated by Greatwall kinase (Gwl), and the resultant pEndos inhibits the phosphatase PP2A-B55, which would otherwise prematurely reverse many CDK-driven phosphorylations. We show here that PP2A-B55 is the enzyme responsible for dephosphorylating pEndos during M phase exit. The kinetic parameters for PP2A-B55's action on pEndos are orders of magnitude lower than those for CDK-phosphorylated substrates, suggesting a simple model for PP2A-B55 regulation that we call inhibition by unfair competition. As the name suggests, during M phase PP2A-B55's attention is diverted to pEndos, which binds much more avidly and is dephosphorylated more slowly than other substrates. When Gwl is inactivated during the M phase-to-interphase transition, the dynamic balance changes: pEndos dephosphorylated by PP2A-B55 cannot be replaced, so the phosphatase can refocus its attention on CDK-phosphorylated substrates. This mechanism explains simultaneously how PP2A-B55 and Gwl together regulate pEndos, and how pEndos controls PP2A-B55.

**\*For correspondence:** mlg11@cornell.edu

**Present address:** †Department of Biological and Chemical Sciences, Wells College, Aurora, United States

## Introduction

Entry of cells into M phase of mitosis or meiosis requires the phosphorylation of thousands of sites on hundreds of proteins (*Dephoure et al., 2008*; *Lindqvist et al., 2009*; *Dulla et al., 2010*; *Olsen et al., 2010*). Many of these sites are substrates of cyclin-dependent kinases (CDKs), most notably MPF (M phase-promoting factor; CDK1-Cyclin B). CDK phosphosites are 'proline-directed', being phosphorylated at serine or threonine followed by proline (*Nigg, 1993*; *Holmes and Solomon, 1996*). These M phase-specific phosphorylations must eventually be removed so that mitotic/meiotic cells can reset to interphase. PP2A-B55 (heterotrimeric protein phosphatase 2A composed of a catalytic C subunit, a structural A subunit, and a B55-type regulatory subunit) is the phosphatase responsible for removing many key CDK-generated phosphorylations at the conclusion of M phase (*Clarke et al., 1993*; *Mayer-Jaekel et al., 1994*; *Mochida and Hunt, 2007*; *Castilho et al., 2009*; *Mochida et al., 2009*; *Schmitz et al., 2010*).

The circuitry's basic logic suggests that PP2A-B55 activity must be downregulated when cells go into M phase; if not, the phosphorylations catalyzed by MPF would be prematurely removed, preventing M phase entry (*Clarke et al., 1993*; *Lee et al., 1994*). Indeed, a regulatory module that turns off PP2A-B55 specifically during M phase has recently been characterized (*Figure 1*). In brief, MPF phosphorylates and thereby activates the Greatwall kinase (Gwl) (*Yu et al., 2006*; *Blake-Hodek et al., 2012*). Activated Gwl phosphorylates small proteins of the Endosulfine family (Endos; [*Gharbi-Ayachi et al., 2010*; *Mochida et al., 2010*]) at a unique, highly conserved site. Gwl-phosphorylated Endos (pEndos) binds to and inactivates PP2A-B55, thus protecting a major class of CDK-governed

**eLife digest** The most dramatic stage of the cell division cycle is M phase, when the cell splits into two genetically identical daughter cells. If this process goes wrong, the cell might die, so cells employ a complicated regulatory process to ensure that M phase begins and ends at the right time.

As with many biological processes, regulation of the cell cycle depends on the activation and inhibition of a range of enzymes. Enzymes act as biological catalysts, binding target molecules (substrates) to active sites so that chemical reactions can take place. However, the activity of the enzyme can be shut down if a different type of molecule, called a competitive inhibitor, binds to the active site.

For M phase to proceed, an enzyme called M phase promoting factor adds phosphate groups to hundreds of target proteins. At the end of M phase, a different enzyme, called PP2A-B55, removes these phosphate groups. Cells can enter M phase because an inhibitor called Endosulfine blocks the active site of the PP2A-B55 enzyme. However, the cells need to unblock the PP2A-B55 enzyme at the end of M phase. Williams et al. have now established the mechanism behind this unblocking of the PP2A-B55 enzyme.

One basis for this mechanism is that Endosulfine works as an inhibitor only when it is phosphorylated (contains a phosphate group). Throughout M phase, a plentiful supply of newly phosphorylated Endosulfine inhibitor molecules binds very tightly to the active sites of the PP2A-B55 enzyme molecules, blocking the enzyme's more loosely binding substrates from accessing the active sites. The second basis for this mechanism is that PP2A-B55 can also slowly remove the phosphate groups from Endosulfine molecules bound at the active site. In other words, phosphorylated Endosulfine works as an inhibitor only because it is really a substrate with special properties. It binds very tightly to the active site, where it is destroyed very slowly. For this reason, Williams et al. have named the process inhibition by unfair competition.

In the final stages of M phase, the cell cannot produce any more phosphorylated Endosulfine molecules, and the PP2A-B55 enzyme can then destroy all the existing inhibitors. Even though this reaction is relatively slow, it is still achieved within a couple of minutes. After its active site is no longer blocked, the PP2A-B55 enzyme is then free to remove the phosphate groups from the target proteins, and M phase can come to an end.

phosphorylations from premature removal during M phase (reviewed in *Glover, 2012*; *Lorca and Castro, 2012*, *2013* and *Hunt, 2013*). The Gwl→pEndos ⊣ PP2A-B55 pathway is critical for M phase entry and maintenance in frog egg extracts, starfish oocytes, and in many cell types in culture or within metazoan organisms (*Archambault et al., 2007*; *Burgess et al., 2010*; *Hara et al., 2012*; *Lorca et al., 2010*; *Voets and Wolthuis, 2010*; *Yu et al., 2004*).

In this report, we explore how this M phase activation system is reversed when cells exit mitosis or meiosis. Not only must CDK-dependent phosphorylations be removed by newly reactivated PP2A-B55, but also Gwl and pEndos must be inactivated at the end of M phase by the removal of their stimulatory phosphorylations (*Figure 1*). If Gwl were to remain active, it would continually keep Endos phosphorylated; PP2A-B55 would stay inactive and the system would remain in M phase because mitotic phosphorylations could not be removed. On the other hand, if Gwl alone were inactivated but pEndos were to stay phosphorylated, M phase would again be maintained. We focus here on the second part of this equation by identifying the phosphatase that dephosphorylates and turns off pEndos.

At the onset of these investigations, we thought that pEndos inactivation could not be achieved solely by PP2A-B55. Not only is the Gwl-phosphorylated pEndos site not proline-directed, as are CDK targets, but also a system based only on PP2A-B55 would be closed and futile: if PP2A-B55 is inactive during M phase, how could this 'dead' enzyme turn itself back on? Clearly, the 'anti-Endos' phosphatase(s) targeting Gwl-phosphorylated pEndos must instead be active during at least late M phase. As we will show, the expectation that anti-Endos is active during M phase has been borne out. However, counterintuitively, we found that almost all of this anti-Endos activity is contributed by PP2A-B55. This surprising conclusion is made possible because the kinetic parameters of this enzyme for pEndos and for CDK-phosphorylated substrates are very different. Based on this divergence, we suggest a straightforward mechanism we call 'inhibition by unfair competition' that explains

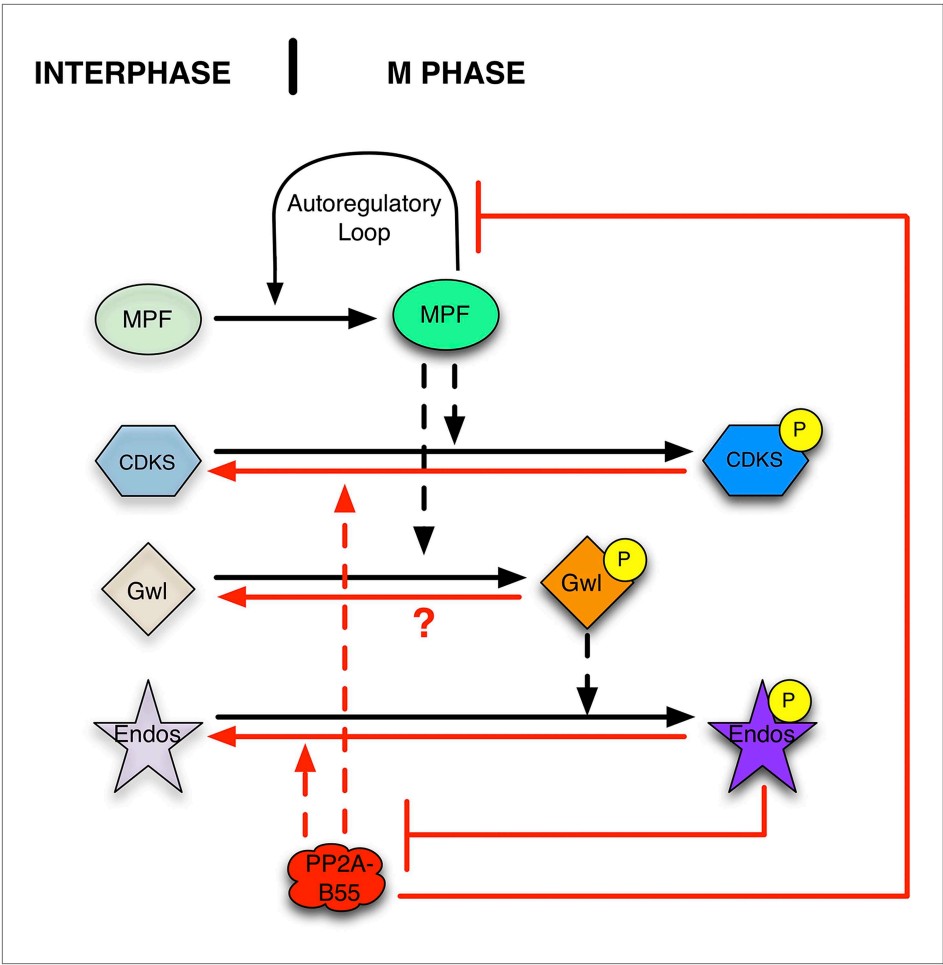

**Figure 1**. Function of the Gwl → pEndos ⊣ PP2A-B55 module in cell cycle transitions. The major driver for transitions between interphase and M phase is the cyclic activation and degradation of MPF (Cdk1-Cyclin B). When activated (in part through a feed-forward autoregulatory loop involving the kinases Myt1 and Wee1 and the phosphatase Cdc25; not shown), MPF phosphorylates many substrates (CDKSs) that play key roles in M phase events. One such MPF substrate is the kinase Greatwall (Gwl), which in its active form phosphorylates Endosulfine (Endos). Phosphorylated Endos binds to and inhibits the phosphatase PP2A-B55. This inhibition protects MPF substrates, including components of the autoregulatory loop, from premature dephosphorylation during M phase entry. During M phase exit, MPF is inactivated by degradation of its Cyclin B component (not shown), and PP2A-B55 becomes reactivated to dephosphorylate many MPF substrates. M phase exit also requires the dephosphorylation and inactivation of both Gwl and Endos. Here, we show that PP2A-B55 catalyzes Endos dephosphorylation. The activities responsible for the inactivation of Gwl currently remain unknown.

how pEndos acts both as an inhibitor and a substrate of PP2A-B55, and how this phosphatase can simultaneously be 'on' for some substrates but 'off' for others. This mechanism may have broad implications for the control of other cellular protein phosphatases.

## Results

As a first step in identifying the phosphatase(s) that dephosphorylate and inactivate pEndos, we assayed various cellular extracts for the release of $^{32}$P from recombinant pEndos previously phosphorylated by Gwl. The extracts' anti-Endos activities were compared with their phosphatase activities against other representative substrates used as internal controls. Most often, we used the recombinant polypeptide CDKS (CDK substrate) phosphorylated in vitro by CDK1-Cyclin A ('Materials and methods'). Previous studies have shown that dephosphorylation at this site is specifically catalyzed by PP2A-B55 (*Mochida and Hunt, 2007*; *Castilho et al., 2009*); hereafter we reference this

dephosphorylation as the 'anti-CDKS' activity. We obtained very similar results in several experiments using a different CDK substrate that includes Fizzy (Fzy) phosphorylated at Ser50, which is also a specific target of PP2A-B55 (*Mochida and Hunt, 2007*; *Castilho et al., 2009*; *Mochida et al., 2009, 2010*).

## Anti-Endos remains active during M phase

We initiated our search for the anti-Endos phosphatase by asking if this activity has the basic characteristic predicted by the logic of the system: in contrast with the anti-CDKS activity of PP2A-B55, which is off during M phase, the anti-Endos phosphatase should remain active during M phase to permit subsequent M phase exit. This question could be addressed readily using extracts of *Xenopus* eggs, which are prepared in an M phase state but can be induced to exit M phase by addition of $Ca^{2+}$ (*Murray and Kirschner, 1989*; *Murray, 1991*; *Tunquist and Maller, 2003*). *Figure 2A* shows that in accordance with this prediction, considerable anti-Endos activity is indeed seen during M phase. The level is roughly half that seen in interphase; as will be explained below, we believe this difference results from competition between exogenous radiolabeled pEndos and endogenous unlabeled pEndos present in M phase but not interphase. As expected from previous studies (*Mochida and Hunt, 2007*; *Castilho et al., 2009*), anti-CDKS activity (i.e., PP2A-B55) was completely blocked in M phase extracts and strongly induced by treatment with $Ca^{2+}$ (*Figure 2A*).

## The predominant anti-Endos activity is associated with PP2A, PP4, or PP6

We next characterized the sensitivity of the anti-Endos phosphatase in concentrated extracts (from *Xenopus* eggs and human, fly, or mouse tissue culture cells) to common phosphatase inhibitors. The properties of anti-Endos studied in all of these extracts were nearly interchangeable. All of the activity in all extracts tested was suppressed by relatively low doses of okadaic acid or calyculin A (*Figure 2B*, *Figure 2—figure supplement 1*), but was completely resistant to the calcineurin (PP2B) inhibitor cyclosporin A (data not shown). These results indicate that the enzyme(s) targeting the Gwl site in Endos belong to the PPP family of phospho-serine/threonine protein phosphatases, which include PP1, PP2A, PP4, PP5, and PP6 (*Swingle et al., 2007*).

PP1 and PP5 are ~10,000-fold more resistant to the inhibitor fostriecin than the PP2A/PP4/PP6 group of enzymes (*Swingle et al., 2007*). In all extracts examined, the majority of anti-Endos activity was sensitive to the same doses of fostriecin that inhibit PP2A-B55's anti-CDKS activity (*Figure 2C*, *Figure 2—figure supplement 2*). Anti-Endos and anti-CDKS activities were both considerably more sensitive to fostriecin than were the dephosphorylations of two other substrates, CDK-phosphorylated histone H1v1.0, which is substantially targeted by PP1-like enzymes (*Paulson et al., 1994*; *Qian et al., 2011*), and histone H3, which is apparently a substrate for both a fostriecin-sensitive and a fostriecin-resistant phosphatase. The predominant anti-Endos activity in many cell types (~70–90% of the total depending on the experiment) is thus due to PP2A, PP4, or PP6. Because the major anti-Endos activity displays closely similar sensitivities to okadaic acid or fostriecin when comparing M phase and interphase frog egg extracts, it appears that the same phosphatase is responsible for this activity during both cell cycle stages (*Figure 2—figure supplements 1C and 2A*).

A minor fraction of anti-Endos is nonetheless fostriecin-resistant; this part of the activity is labile and is seen in some extract preparations (*Figure 2—figure supplement 2A–C,F*) but not others (*Figure 2C*, *Figure 2—figure supplement 2D,E*). This secondary activity, possibly due to some form of PP1, is likely responsible for the modest okadaic acid resistance of anti-Endos (relative to anti-CDKS) seen in concentrated *Xenopus* extracts (*Figure 2—figure supplement 1C*), where the proportion of fostriecin-resistant activity is highest (*Figure 2—figure supplement 2A,B*). In the 'Discussion', we argue that this minor component of anti-Endos is unlikely to be of great physiological importance; in the remainder of the 'Results', we thus focus on the predominant fostriecin-sensitive enzyme.

## Anti-Endos and anti-CDKS phosphatase activities differ in many properties

In spite of the fact that anti-Endos and anti-CDKS are both associated with the same highly related subfamily of PPP phosphatases including PP2A, PP4, and PP6, further experiments revealed clear divergences in the behavior of these two phosphatase activities.

First, we characterized the response of anti-Endos and anti-CDKS activities to tautomycetin and its relative tautomycin. These drugs have been reported to be much more effective as inhibitors of PP1 than PP2A (*Gupta et al., 1997*; *Mitsuhashi et al., 2001*; *Kelker et al., 2009*). This conclusion

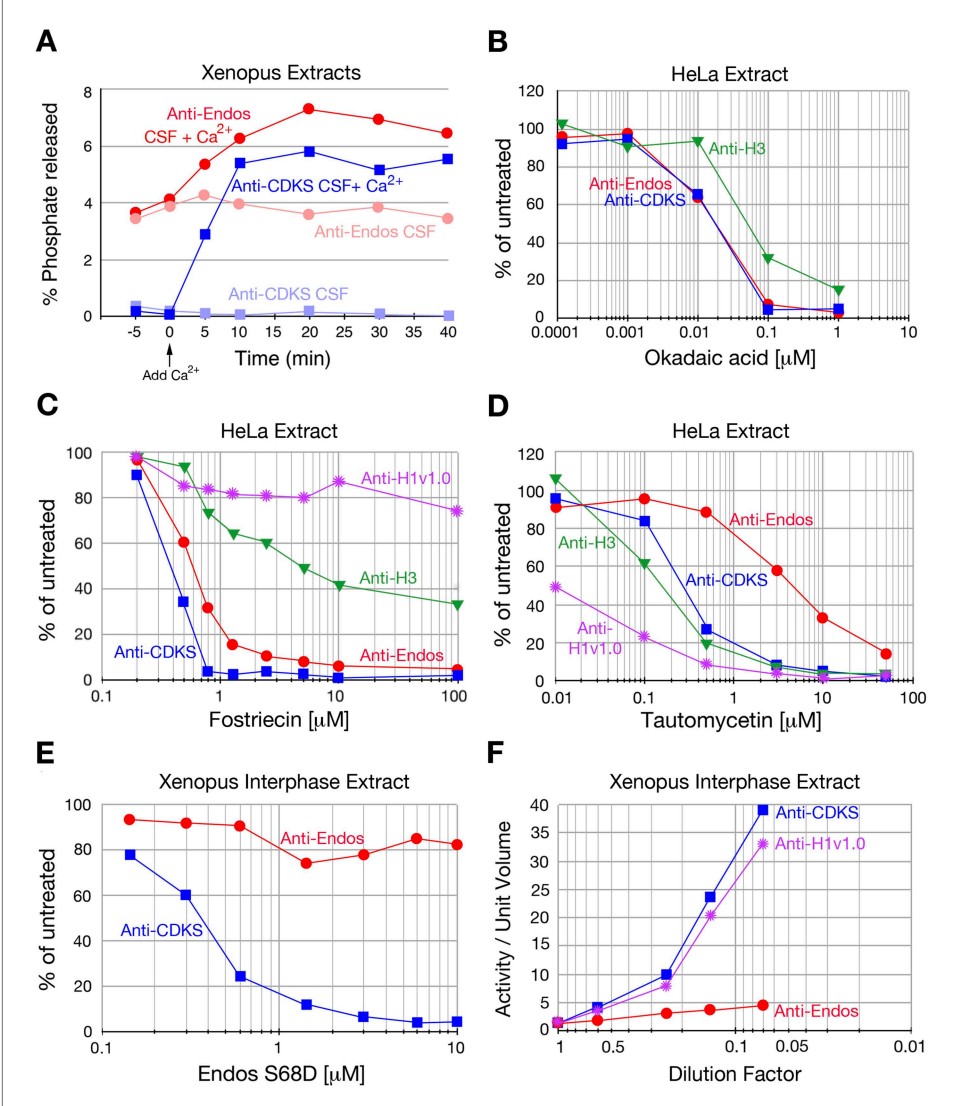

**Figure 2**. Characterization of anti-Endos in extracts. In all parts of this figure, red circles depict anti-Endos, whereas blue squares represent anti-CDKS. (**A**) Anti-Endos is present during M phase. *Xenopus* CSF (M phase) extracts were incubated at 22°C. At time t = 0, $Ca^{2+}$ was added to half of the extract to induce M phase exit; control extract without $Ca^{2+}$ remained in M phase. At the indicated times, aliquots were assayed for anti-Endos and anti-CDKS as described in 'Materials and methods'. During M phase, anti-CDKS (light blue squares) is undetectable, whereas anti-Endos (light red circles) is active. As the extracts exit M phase (interphase is achieved within 15–20 min of $Ca^{2+}$ addition; [**Yu et al., 2006**; **Zhao et al., 2008**; **Castilho et al., 2009**]), anti-CDKS activity (dark blue squares) is strongly induced, while anti-Endos (dark red circles) increases about twofold. (**B**–**E**) Drug sensitivities of phosphatase activities. Y-axis values represent the percentage of the phosphatase activity for the given combination of extract and substrate measured in the absence of the inhibitor. Anti-Endos and anti-CDKS have similar sensitivities to okadaic acid and fostriecin, but anti-Endos is substantially more resistant than anti-CDKS to tautomycetin and phosphomimetic Endos S68D. In **B** and **C**, green triangles represent dephosphorylation activity against CDK-phosphorylated Histone H3; in **C**, purple stars are activity against CDK-phosphorylated Histone H1v1.0. In part **C**, the fostriecin resistant portions of the H3 phosphatase (about 40% of the total) and the H1v1.0 phosphatase (about 80% of the total) likely represent PP1 activity. The HeLa extracts examined in panels **B**–**D** were from asynchronous cells, the vast majority of which are in interphase. (**F**) The specific activities of anti-CDKS and anti-H3 increase upon dilution of the extract, presumably because weakly binding inhibitors are titrated away, but the specific activity of anti-Endos increases at most only marginally upon dilution. The y-axis shows the phosphatase activity on the indicated substrates, normalized to the original volume of undiluted extract. In all

*Figure 2. Continued on next page*

*Figure 2. Continued*

panels, *n* = 1; biological and evolutionary replicates of the experiments in panels **B–D** are presented in *Figure 2 figure supplements 1–5*.

The following figure supplements are available for figure 2:

**Figure supplement 1**. Anti-Endos is completely inhibited by okadaic acid and calyculin.

**Figure supplement 2**. Anti-Endos is mostly fostriecin-sensitive.

**Figure supplement 3**. Anti-Endos is more tautomycetin/tautomycin-resistant than anti-CDKS.

**Figure supplement 4**. Anti-Endos is resistant to phosphomimetic Endos.

**Figure supplement 5**. The specific activity of anti-Endos does not increase upon substrate dilution.

**Figure supplement 6**. Estimating relative levels of Twins (B55) and Endos proteins in *Drosophila* S2 cells.

---

was verified in our own system using purified enzymes (*Figure 2—figure supplement 3A*) and by the tautomycetin sensitivity of phosphatase activity in extracts against the PP1-specific substrate Histone H1v1.0 (*Figure 2D*). In contrast, the anti-Endos activity in all extracts tested is 6- to 10-fold more resistant to tautomycetin than is anti-CDKS (*Figure 2D*, *Figure 2—figure supplement 3B–F*). This result is surprising because we had not anticipated the existence of a PPP family phosphatase more tautomycetin-resistant than PP2A-B55. The tautomycetin/tautomycin resistance of anti-Endos is due to the major, fostriecin-sensitive component, because extracts lacking the labile fostriecin-resistant enzyme are still resistant to tautomycetin (the extracts used in *Figure 2C* and *Figure 2D* are identical).

Second, we found that in contrast to anti-CDKS, anti-Endos is relatively unaffected by the presence of constitutively activated Endos. Endosulfine thiophosphorylated by Gwl acts in vitro as a specific inhibitor of PP2A-B55's anti-CDKS activity; this effect is also observed using Endos bearing a phosphomimetic mutation of the Gwl target site (*Gharbi-Ayachi et al., 2010*; *Mochida et al., 2010*; *Kim et al., 2012*). However, the anti-Endos activity in extracts is substantially resistant to inhibition by phosphomimetic *Drosophila* Endos (S68D) (*Figure 2E*, *Figure 2—figure supplement 4*).

Finally, it has previously been reported that the dilution of cellular extracts leads to large increases in the specific activity of protein phosphatases, possibly caused by the decreased association, due to mass action, of the phosphatase with weak inhibitors in the extracts (*Cohen, 1989*; *Mochida et al., 2009*). We have verified this dilution effect for anti-CDKS and anti-H1v1.0, but in contrast, anti-Endos activity is very much less affected by extract dilution (*Figure 2F*, *Figure 2—figure supplement 5*).

The data to this point suggest that the major anti-Endos enzyme is some form of PP2A, PP4, or PP6. As we anticipated, anti-Endos is clearly differentiated from PP2A-B55 (anti-CDKS) in terms of its constitutive activity during M phase. Anti-Endos and anti-CDKS also diverge strongly in terms of their relative sensitivities to tautomycetin, phosphomimetic Endos, and extract dilution.

## Depletion or mutation of PP2A-B55 removes most anti-Endos activity

To pinpoint which of the candidate phosphatases targets the Gwl phosphosite on pEndos, we examined anti-Endos activity in extracts of cultured cells depleted for phosphatase subunits (*Figure 3*). Treatment of *Drosophila* S2 cells with dsRNAs for the single PP2A catalytic subunit gene in the fly genome (called *mts*) caused the loss of 70–75% of the anti-Endos activity (*Figure 3A*). Success of the RNAi was verified by Western blot (*Figure 3B*), and by the almost complete loss of anti-CDKS activity in the dsRNA-treated cells (*Figure 3A*). Depletion of the A (structural) subunit of PP2A also caused substantial loss of anti-Endos activity (*Figure 3*); as expected from previous studies (*Silverstein et al., 2002*), RNAi for fly PP2A-A destabilizes PP2A-C, and vice versa. In contrast, RNAi for *PP1-87B*, whose product accounts for more than 80% of the PP1 activity in *Drosophila* larvae (*Dombradi et al., 1990*), only slightly decreased anti-Endos (*Figure 3*). dsRNAs against coding sequences for all other PPP family catalytic subunits (PP4, PP5, and PP6) had no obvious effects on anti-Endos levels (data not shown).

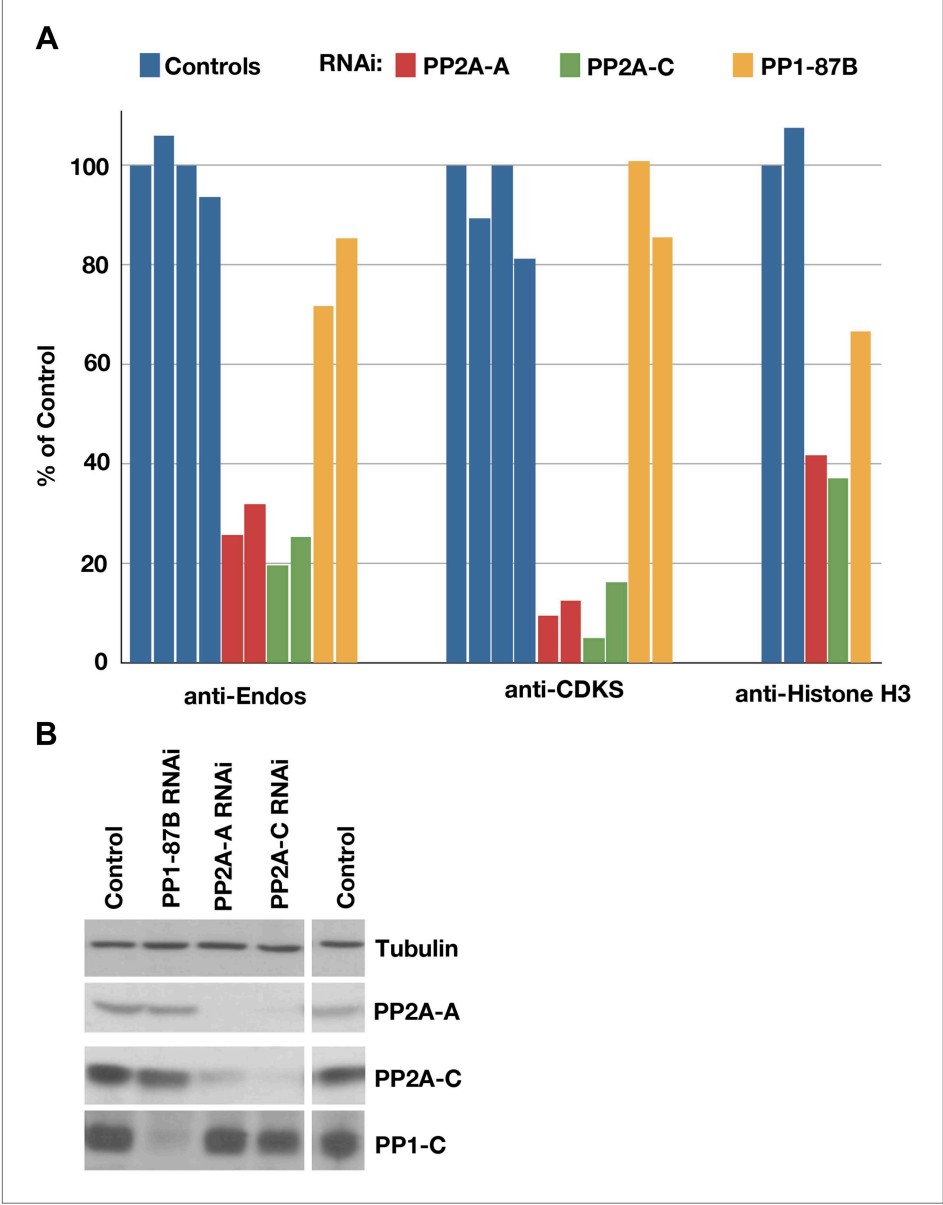

**Figure 3**. Depletion of PP2A by RNAi disrupts a major component of anti-Endos activity. (**A**) Phosphatase assays for anti-Endos, anti-CDKS, and anti-Histone H3 activities in mock-treated control *Drosophila* S2 tissue culture cells (blue), and S2 cells treated with dsRNAs for the PP2A-A subunit (red; the gene is *PP2A-29B*), the PP2A-C subunit (green; the gene is *mts*), and the major PP1-C subunit (yellow; the gene is *PP1-87B*, which accounts for more than 80% of PP1-related activity against generic substrates [***Dombradi et al., 1990***]). Each column represents a separate biological replicate (*n* = 4 for the controls and *n* = 2 for each of the three RNAi treatments); values were normalized for total protein concentrations in the extracts. One of the four replicates of the control assay was arbitrarily chosen as the 100% reference. The data indicate that a form of PP2A is responsible for about 90% of anti-CDKS, ~70 to 80% of anti-Endos, and ~60% of anti-H3; the residual activities are likely due mostly to forms of PP1, although PP5 may also contribute in the case of anti-H3 (**Figure 4**). (**B**) Western blots showing effectiveness of the RNAi treatments. As reported elsewhere (***Kamibayashi et al., 1992***), the stabilities of the PP2A-A and PP2A-C subunits are mutually interdependent.

The following figure supplements are available for figure 3:

**Figure supplement 1**. Neither PP4 nor PP5 is a major contributor to anti-Endos.

Analogous experiments to deplete phosphatase catalytic subunits from HeLa cells using siRNAs were clearly successful only in the case of PP4, where no effects on either anti-Endos or anti-CDKS activities were observed (*Figure 3—figure supplement 1A,B*). Mouse MEF cells homozygous for a knockout allele of the *PP5* gene (*Yong et al., 2007*) also displayed the same levels of anti-Endos and anti-CDKS activities as control cells (*Figure 3—figure supplement 1C,D*).

These knockdown/mutation experiments argue strongly that the predominant anti-Endos phosphatase is some form of PP2A. Most PP2A activity in vivo is ascribed to heterotrimeric enzymes containing the C and A subunits plus one of several kinds of regulatory B subunits (*Janssens et al., 2008*). To distinguish which form of PP2A is anti-Endos, we took advantage of the fact that the *Drosophila* genome has a near-minimal set of genes encoding PP2A regulatory subunits (e.g., one B55-type subunit as opposed to four in mammals; and two B56-class [B'] subunits vs five in mammals). Null or strongly hypomorphic mutations are available for almost all of these fly genes (*Uemura et al., 1993*; *Mayer-Jaekel et al., 1994*; *Chen et al., 2002*; *Hannus et al., 2002*; *Viquez et al., 2006*); animals homozygous for these mutations usually survive until third instar larval or pupal stages because maternal contributions are sufficient for earlier stages of development (*Gatti and Goldberg, 1991*).

Extracts prepared from mutant third instar larvae were assayed for anti-Endos and anti-CDKS activities, as well as phosphatase activity against Histone H1v1.0 as an additional control. The results were striking and unanticipated: Relative to wild-type, larvae homozygous for mutations in *twins* (the gene encoding the sole B55-type subunit in flies) were not only deficient in anti-CDKS as expected, but they also lacked anti-Endos; the same was true for brains isolated from these larvae (*Figure 4A*). Activities against Histone H1v1.0 were virtually unaffected in *twins* mutants. On Western blots, the *twins* larvae had no detectable Twins (B55) protein; moreover, gross levels of PP2A-A and PP2A-C were not diminished by the absence of Twins (*Figure 4B*). None of these phosphatase activities were altered in larvae carrying mutations in genes for any other tested PP2A regulatory subunit, for the PPP4R3 regulatory subunit of PP4 (*flfl*), or for any of the PP1 catalytic subunits (with the possible exception of a decrease in Histone H1v1.0 dephosphorylation in *PP1-87B* larvae; *Figure 4A*).

## The predominant anti-Endos activity co-purifies with the PP2A-B55 heterotrimer

As just seen, depletion/mutation of any of the three components of the PP2A-B55 heterotrimer removes the large majority of the phosphatase activity directed against the Gwl site in pEndos. These findings were very surprising, given that PP2A-B55's action against CDK-phosphorylated substrates differs in many properties from anti-Endos, and the sequence motifs phosphorylated by Gwl and CDKs are very different. We were thus concerned that the effects of PP2A-B55 depletion on pEndos dephosphorylation could be indirect: that is, PP2A-B55 might not be anti-Endos, but its loss might instead disrupt another phosphatase that does play such a role.

To address whether anti-Endos is indeed a PP2A-B55 heterotrimer, we fractionated asynchronous HeLa cell extracts by Mono-Q chromatography (*Figure 5*). Assays of the fractions revealed that anti-Endos and anti-CDKS activities co-purified. The peak activities towards both substrates coincided with the fractions containing the three components of PP2A-B55 (the A, B55, and C subunits), although the PP1 catalytic subunit was also found in these same fractions. PP5 and PP6 elute from the Mono-Q column at much lower salt concentrations, while PP4 and the B56 and B''' (striatin) regulatory subunits of PP2A elute at higher salt than the peak anti-Endos/anti-CDKS (*Figure 5*). The anti-Endos activity in the peak fractions has the same characteristics as in concentrated extracts: it is sensitive to okadaic acid and fostriecin, but insensitive to tautomycetin and phosphomimetic *Drosophila* Endos (data not shown). The fractionation results are consistent with the idea that anti-Endos is indeed the PP2A-B55 heterotrimer and further exclude most other PPP enzymes as candidates.

We next purified recombinant PP2A-B55 heterotrimer from lysates of human embryonic kidney (HEK) cells transfected with constructs expressing FLAG-tagged B55α or B55δ isoforms, or the B56β isoform as a control (as described in *Adams and Wadzinski (2007)*). On silver stained SDS-PAGE gels, the FLAG immune complexes showed predominant bands at the positions of the A, FLAG-B55 or FLAG-B56, and C subunits of PP2A (*Figure 6—figure supplement 1*). Some variable contaminating bands were also seen in control immune complexes isolated from lysates of cells transfected with empty vectors. Mass spectrometry identified these proteins as keratins or immunoglobulin chains (from the antibody on the FLAG affinity beads) unlikely to contribute to phosphatase activities.

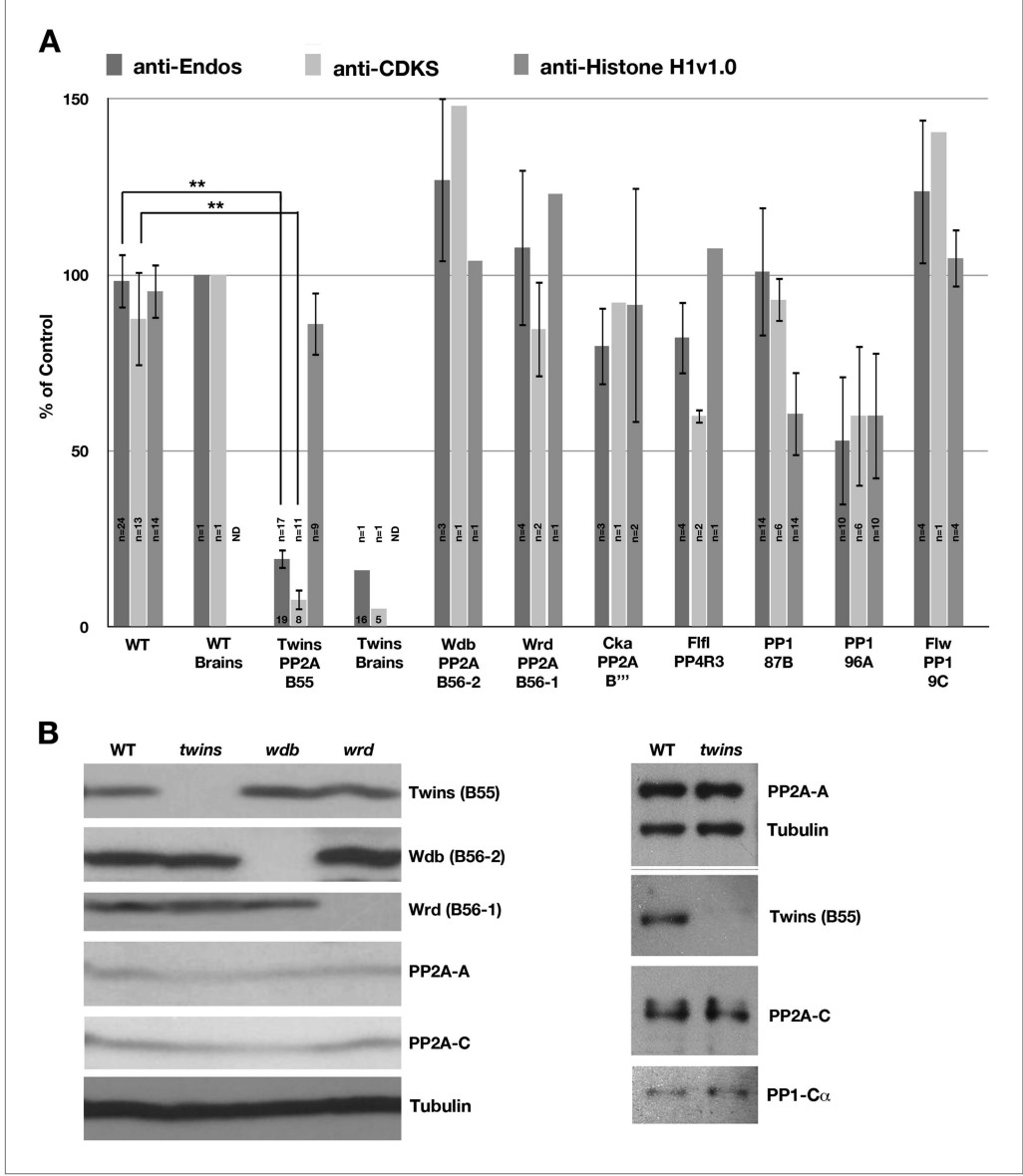

**Figure 4**. *Drosophila* larvae lacking B55 are deficient in anti-Endos. (**A**) Extracts were prepared from larvae with null or strong loss-of-function alleles of the indicated genes, and assayed for anti-Endos, anti-CDKS, and anti-Histone H1v1.0 activities. Values were normalized for the total protein concentrations in the extracts; one replicate of the control assay (on a wild-type larva) was arbitrarily chosen as the 100% reference. Details about the genotypes involved are given in 'Materials and methods'. The number of biological replicates for each genotype is presented inside the corresponding bar, with standard deviations shown. Levels of anti-Endos are much lower in *twins* (encoding the sole B55 subunit in *Drosophila*) larvae than in wild-type (WT) controls ($p < 10^{-20}$; Student's two-tailed *t* test); this is also the case as expected for anti-CDKS ($p < 10^{10}$). As a control, phosphatase activity against Histone H1v1.10 is relatively unaffected in *twins* mutant larvae. Anti-Endos and Anti-CDKS activities were also severely compromised in extracts made from brains isolated from *twins* mutant animals. No consistent effects were observed in extracts made from larvae mutant for genes encoding the other phosphatase subunits indicated, with two exceptions. First, extracts from larvae mutant for *PP1-96A* displayed only about half the level of phosphatase activities measured with all three substrates. This is probably a systematic error caused by the *ebony* mutation in these animals, producing a dark pigment that caused an overestimation of the amount of total protein concentration. Second, the lower level of activity against Histone H1v1.0 in animals mutant for *PP1-87B* (the predominant PP1 catalytic subunit in *Drosophila*; **Dombradi et al., 1990**) is likely caused by the targeting of this substrate by the PP1-87B phosphatase. (**B**) Western blots of extracts from larvae mutant for genes encoding B55 (*twins*) and B56 (*widerborst* [*wdb*] and *well-rounded* [*wrd*]) regulatory subunits of PP2A. The blot at the right verifies that in *twins* mutants, the B55 protein is missing, while the PP2A-A and PP2A-C subunits are present in normal amounts.

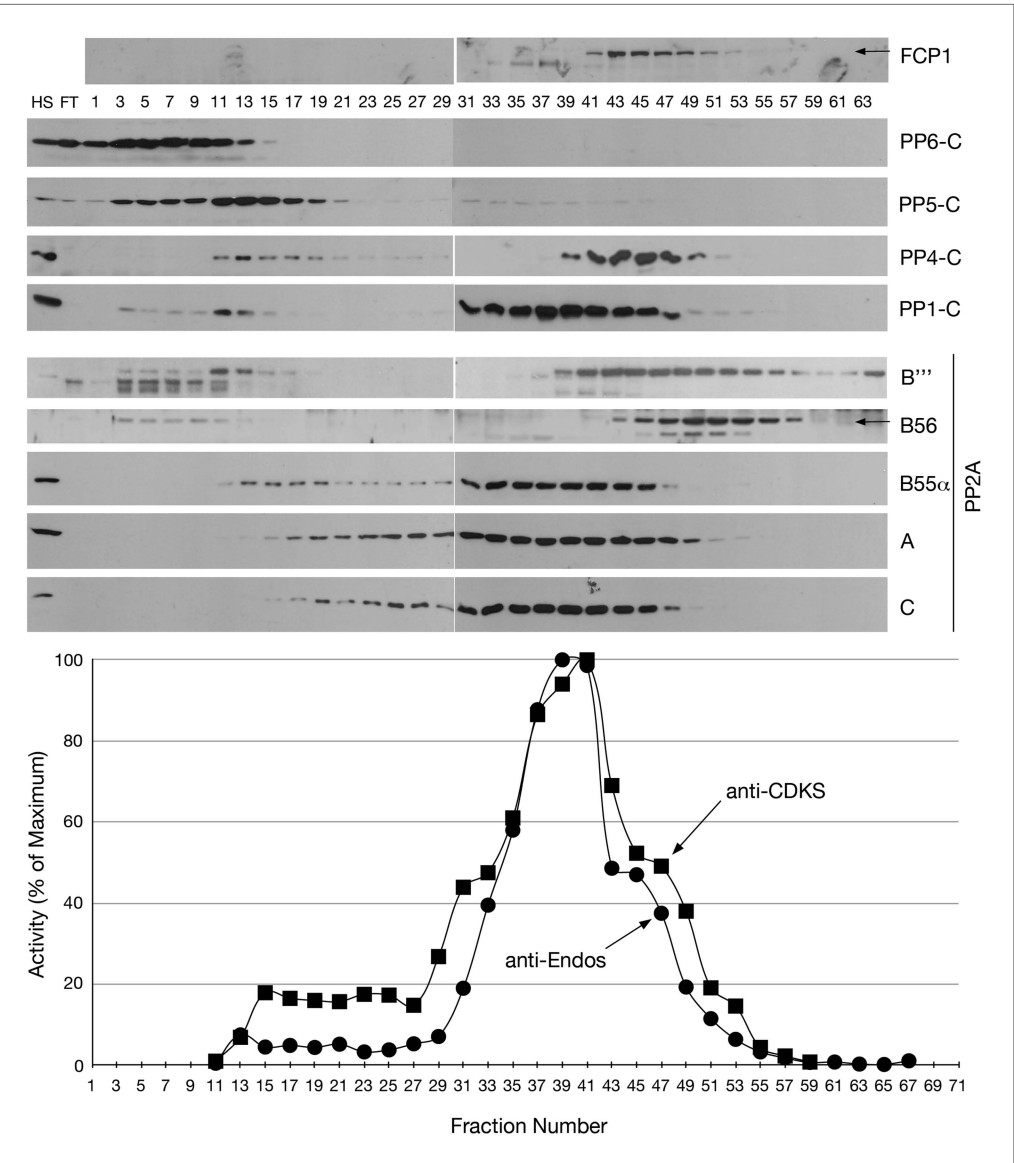

**Figure 5**. Mono-Q chromatography of HeLa extracts. Total extract from asynchronous HeLa cells (high speed supernatant; HS) was applied to the column; FT is the flow-through. Proteins were eluted with linear gradient from 150 mM (Fraction 1) to 500 mM (Fraction 75) of NaCl. Individual fractions were assayed (*n* = 1 assay per data point) for anti-Endos (black circles) and anti-CDKS (black squares), and were also examined for the indicated phosphatase subunits by Western blot.

Heterotrimers containing B55α and B55δ isoforms efficiently dephosphorylated pEndos and pCDKS substrates (*Figure 6A*). Pilot experiments allowed us to choose concentrations of the enzymes and time of reactions in which subsequent investigations could be performed in the linear range for these variables (*Figure 6—figure supplement 2*). Neither control PP2A-B56β heterotrimers nor empty vector preparations made in parallel displayed any activity against either substrate. Neither the anti-Endos nor anti-CDKS functions of the PP2A-B55 preparations could be attributed to heterotrimer dissociation into A–C dimers, because purified dimer does not target either substrate even when in 10-fold excess relative to the heterotrimeric holoenzymes (*Figure 6A*). The anti-Endos and anti-CDKS activities of the PP2A-B55 heterotrimer were similarly sensitive to fostriecin, as expected if the catalytic subunit is the same (*Figure 6B*). However, the anti-Endos activity of PP2A-B55 was much less sensitive to either tautomycetin or phosphomimetic Endos than was the anti-CDKS activity of the same

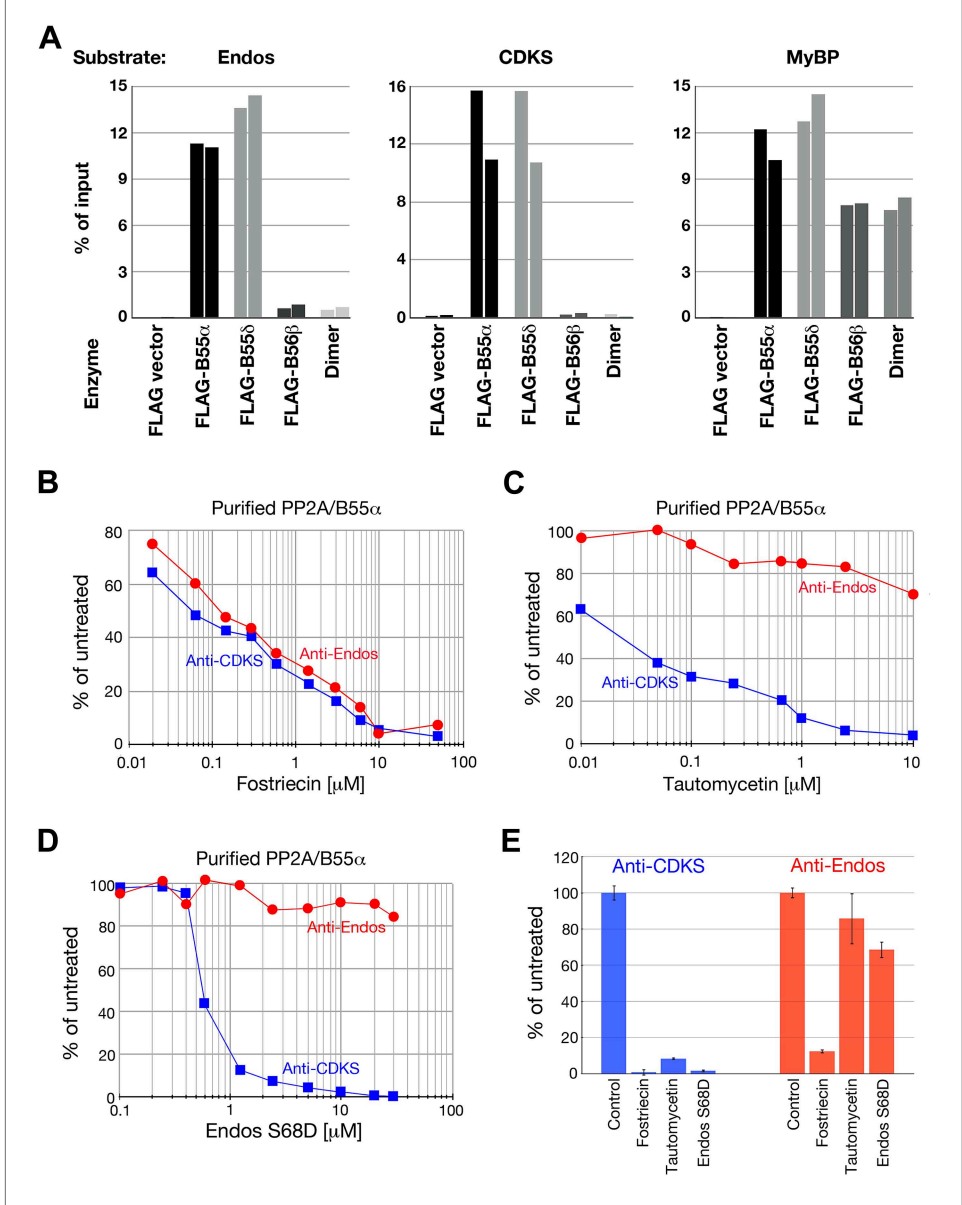

**Figure 6**. Anti-Endos activity of purified PP2A-B55. (**A**) PP2A-B55 heterotrimers efficiently dephosphorylate Gwl-phosphorylated Endos. Each column represents an individual experiment (a biological replicate) in which the indicated enzyme preparations shown in *Figure 6—figure supplement 1* (normalized for the amount of PP2A-C subunit where possible, or for the volume of transfected cells in the case of the vector-only control preparation) were assayed for anti-Endos activity, anti-CDKS activity, or generic activity against myelin basic protein (MyBP) phosphorylated with PKA. The y-axis indicates the percentage of radioactivity in the input substrate that was freed by enzyme treatment. The control preparation had no activity against any of these substrates, whereas PP2A-B55α and PP2A-B55δ heterotrimers showed similar strong levels of activity against all three substrates. PP2A-B56β heterotrimers and PP2A-A/C heterodimers, both of which dephosphorylated the generic MyBP substrate, failed to dephosphorylate either Gwl-phosphorylated pEndos or the pCDKS substrate. (**B–D**) Dose-response curves to phosphatase inhibitors (fostriecin, tautomycetin, and phosphomimetic Endos S68D as indicated) for the anti-Endos (red circles) and anti-CDKS (blue squares) activities of purified PP2A-B55α heterotrimers. Each point represents a single assay (*n* = 1). The anti-Endos activity of purified PP2A-B55α heterotrimers has the same characteristics as the predominant anti-Endos activity in whole cell extracts. In **D**, the fostriecin used has lost potency during the ~6 months of storage after the experiments shown in *Figure 2* were performed; fostriecin is well known to be somewhat labile in this time frame (*Weiser et al., 2003*). It is nonetheless clear that both the anti-Endos and anti-CDKS functions of
*Figure 6. Continued on next page*

*Figure 6. Continued*

PP2A-B555α display similar dose responses to this drug. (**E**) Technical replicates (*n* = 3) of untreated purified PP2A-B55 and enzyme treated with the maximal doses of the inhibitors shown in panels **B**–**D** (20 µM fostriecin, 10 µM tautomycetin, and 20 µM Endos S68D) were performed to assess reproducibility; symbols indicate average values and bars representing one standard deviation are shown.

The following figure supplements are available for figure 6:

**Figure supplement 1**. Preparations of PP2A heterodimer and heterotrimers.

**Figure supplement 2**. Characterization of phosphatase assays using purified PP2A-B55.

---

preparations (*Figure 6C,D*), as was also true for unfractionated extracts (review *Figure 2D,E*) and the peak Mono-Q fractions (data not shown).

## pEndos is a tight-binding but slowly-dephosphorylated substrate of the PP2A-B55 heterotrimer

To understand the different properties of PP2A-B55 with respect to pEndos and CDK-phosphorylated substrates, we performed kinetic analyses of phosphatase activities using purified components and analyzed the data using the Michaelis–Menten reaction scheme allowing for tight-binding ('Materials and methods'). The results are summarized in *Table 1*; the data on which these results were based are presented in *Figure 7*. Measurements of nanomolar or subnanomolar $K_m$ values for the dephosphorylation of pEndos by traditional plots of initial reaction rates as a function of the substrate concentration are technically challenging because of the very low concentrations of enzyme and substrate involved; these issues could lead to overestimates of the $K_m$. We therefore supplemented these studies (*Figure 7A–C*) with a more accurate alternative approach based upon competition between pEndos and okadaic acid for access to the enzyme's active site ('Materials and methods'; *Figure 7D,E*).

The data summarized in *Table 1* indicate that the $k_{cat}$ for the dephosphorylation of pEndos by PP2A-B55 is two to three orders of magnitude slower than that for the reaction of the same enzyme with pCDKS substrate, while the $K_m$ values for the two substrates differ by more than four and perhaps even five orders of magnitude. *Table 1* presents the results as a range of values representing multiple experiments with several independent preparations of PP2A-B55 and either pEndos or pCDKS. We believe the actual $k_{cat}$ for the pEndos reaction is likely to lie in the higher end of its range, because some measurements were taken with older preparations of enzyme that had lost some activity. We further anticipate that the actual $K_m$ for this same reaction is in the lower end of its range, because we obtained lower values using hexahistidine-tagged pEndos than with a pEndos substrate containing a

---

**Table 1.** Kinetic parameters of dephosphorylation by PP2A-B55

| Substrate | $K_m$ (µM) | $k_{cat}$ (sec$^{-1}$) | n* | Method† | Representative experiment |
|---|---|---|---|---|---|
| CDKS | 71–99 | 21–25 | 2 | *v* vs [*S*] | *Figure 7A* |
| Fzy Ser50 | >100‡ | >15‡ | 1 | *v* vs [*S*] | *Figure 7A* |
| pEndos | 0.0009–0.0017 | 0.021–0.035 | 2 | *v* vs [*S*] | *Figure 7B,C* |
| | | 0.018–0.066§ | 16§ | | |
| | 0.0004–0.0011 | 0.005–0.037 | 3 | OA competition | *Figure 7D,E* |

*Number of independent experiments; each point in each experiment included 3–4 measurements.
†*v* vs [*S*] experiments measured the initial rate of reaction as a function of substrate concentration; OA competition experiments involved measurements of pEndos dephosphorylation as a function of okadaic acid competition.
‡Because sufficiently high concentrations of the Fzy Ser50 substrate were not obtained, non-linear regression analysis was inherently inaccurate in determining kinetic parameters. The values given are conservative lower limits.
§The specific activities of multiple independent preparations of purified PP2A-B55 were measured at high pEndos substrate concentrations (5 µM) approximately 1000-fold in excess of the $K_m$.

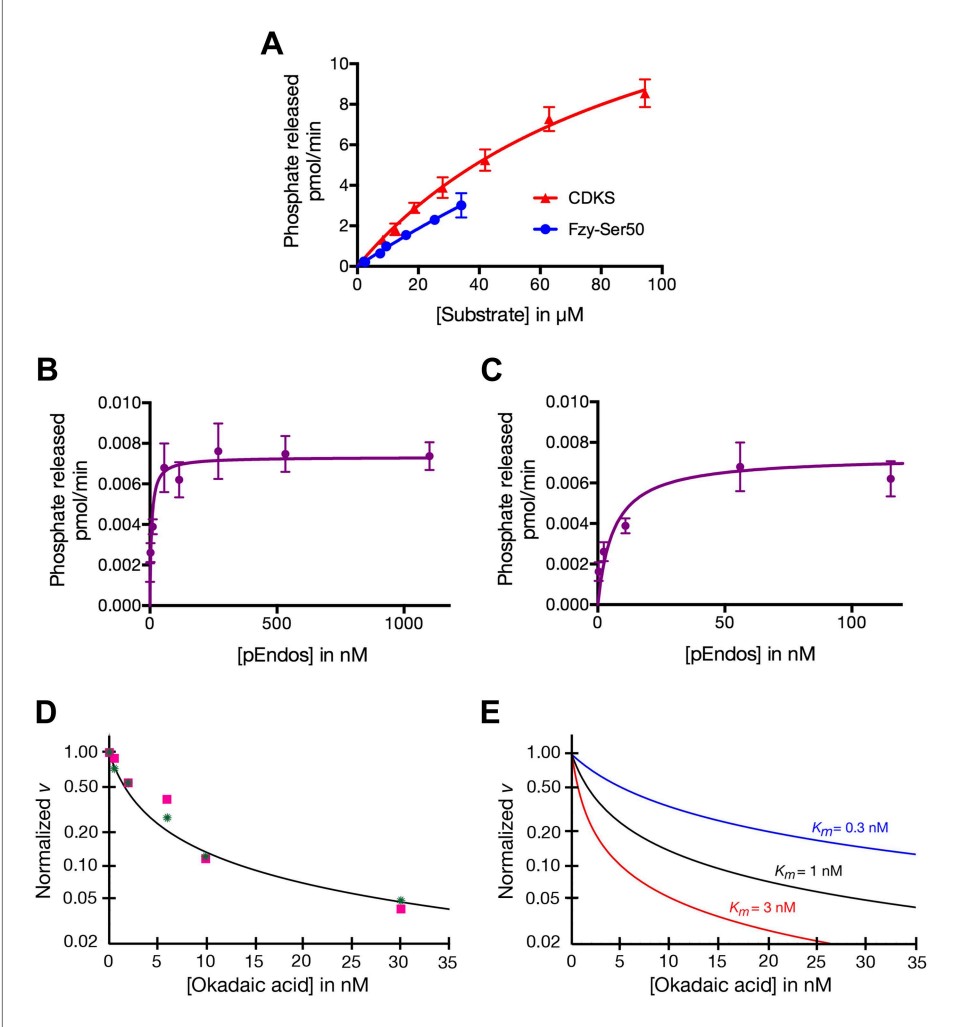

**Figure 7**. Determinations of the kinetic parameters shown in *Table 1*. (**A**) Determination of kinetic parameters for the dephosphorylation of CDKS and Fzy-Ser50 substrates. One representative experiment of each is shown. Initial rates of dephosphosphorylation were determined with increasing substrate concentrations; the concentration of purified PP2A-B55 was 1 nM. The amount of [32]P phosphate released was in all samples less than 20% of the input. For pCDKS substrate, $n = 4$; for pFzy-Ser50 substrate, $n = 3$ (technical replicates). The data were analyzed by non-linear regression analysis (using Prism 6 software) to determine $K_m$ and $k_{cat}$ values incorporated into the data shown in *Table 1*. Because we could not obtain sufficiently concentrated preparations of the Fzy-Ser50 substrate, the uncertainty in these parameters is very high, but the graphs illustrate that the behaviors of these two substrates are nonetheless much more similar to each other than either is to pEndos. (**B** and **C**) Determination of kinetic parameters for the dephos-phorylation of pEndos from the initial reaction velocity as a function of substrate concentration. One representative experiment is shown. Initial rates of dephosphosphorylation were determined with increasing substrate concentra-tions. The concentration of purified PP2A-B55 was 0.5 nM; $n = 4$ technical replicates at each substrate concentration point. The amount of [32]P phosphate released was in all samples less than 20% of the input. The data were analyzed by non-linear regression analysis as detailed in the 'Materials and methods' to determine $K_m$ and $k_{cat}$ values shown in *Table 1*. **B** and **C** graph the same data at different ranges of pEndos concentration. Note that in **B**, the rate of pEndos dephosphorylation remains constant near $V_{max}$ over a 20-fold range from 50 to 1100 nM, indicating that pEndos does not inhibit its own dephosphorylation. (**D** and **E**) Determination of kinetic parameters for the dephosphorylation of pEndos from competition with okadaic acid. Note the logarithmic scale on the *y*-axes. (**D**) Initial rates of pEndos dephosphorylation were determined in the presence of increasing concentrations of okadaic acid; one representative experiment is shown. The amount of [32]P phosphate released was in all samples less than 20% of the input. Two independent experiments (technical replicates) are indicated with pink squares and green stars. The concentration of pEndos in this reaction was 50 nM, the concentration of purified PP2A-B55 was 0.25 nM, and the concentration of okadaic acid varied as shown. The data were fit by non-linear regression analysis to the mathematical model described in 'Materials and methods'; calibration curves based on this model are shown in (**E**).

bulkier tag (maltose binding protein) that might have slightly impaired the interaction. (To appreciate the difficulties inherent in these measurements, it is informative to consider that, of more than 6200 enzymatic reactions currently cataloged in the BRENDA enzyme database (http://www.brenda-enzymes.org; *Schomburg et al., 2013*), only three have $K_m$ values less than 10 nM and none has a subnanomolar $K_m$.) In any event, the differences between the reactions using pEndos or pCDKS substrates are so pronounced that any variations within the ranges shown in *Table 1* are ultimately inconsequential to the models presented below.

These unusual kinetic properties explain the apparent paradox that a single enzyme, PP2A-B55, exhibits both anti-CDKS and anti-Endos activities that differ markedly in their responses to inhibition by both tautomycetin/tautomycin and an Endos phosphomimetic protein and to the dilution of extracts. In each case, the discrepancy is explained by careful quantitative attention to the relationship between the $K_i$ of the inhibitor and the $K_m$ of the target: roughly speaking, an inhibitor will be effective only if its $K_i$ is less than the $K_m$ for the substrate. Thus, as was seen in *Figure 6C*, tautomycetin, which has an IC$_{50}$ (which provides a very rough estimate of $K_i$) of 62 nM (*Mitsuhashi et al., 2001*), effectively inhibits PP2A-B55 dephosphorylation of pCDKS ($K_m$ ~ 90 µM, *Table 1*), but only weakly inhibits PP2A-B55 dephosphorylation of pEndos ($K_m$ ~ 1 nM, *Table 1*). The same consideration explains the difference in inhibition by the *Drosophila* S68 Endos phosphomimetic protein, whose IC$_{50}$ (approximately 500 nM from *Figure 6D*) is again intermediate between that for dephosphorylation of pEndos and pCDKS. Although the identity of the inhibitor(s) present in extracts that presumably underlie the dilution effects seen in *Figure 2F* are unknown, the different sensitivities of anti-Endos and anti-CDK to extract dilution can be rationalized if the inhibitors' $K_i$ values are intermediate between the $K_m$'s of pEndos and pCDKS.

## pEndos inhibition of PP2A-B55 depends on competition at the active site

Two findings clearly indicate that pEndos interacts with the active site of PP2A-B55. First, pEndos is a substrate that is dephosphorylated by this enzyme at a measurable rate; and second, pEndos competes with okadaic acid, a drug known to bind only to PP2A's active site (*Xing et al., 2006*). Our results provide no indication that pEndos inhibits PP2A-B55 by binding to a second, allosteric site separate from the active site at which it is dephosphorylated. Such allosteric down-regulation would cause the rate of pEndos dephosphorylation to decrease when the substrate is added at very high concentrations, but this is not the case (*Figure 7B*). Furthermore, the different responses of purified PP2A-B55's anti-Endos and anti-CDKS activities to the addition of phosphomimetic *Drosophila* Endos (S68D in *Figure 6D*) also argue strongly against inhibition at an allosteric site. If pEndos inhibited PP2A-B55 by binding to a site other than the active site, phosphomimetic Endos would have been expected to inhibit the dephosphorylations of pEndos and pCDKS at identical concentrations, and this is clearly not the case.

*Figure 8A* provides yet another demonstration that the ability of pEndos to inhibit PP2A-B55 is mostly dependent on interactions at the enzyme's active site. We assayed the response of PP2A-B55's anti-CDKS activity to increasing concentrations of either unphosphorylated Endos or Endos previously thiophosphorylated by Gwl kinase; the incorporated thiophosphate cannot be removed by this enzyme (*Morgan et al., 1976*, and data not shown). The IC$_{50}$ of thiophosphorylated Endos determined in this way was 197 ± 14 pM; by comparison, the IC$_{50}$ of nonphosphorylated Endos was measured to be 566 ± 65 nM. Thus, the (thio)phosphorylation of Endos increases its binding affinity for PP2A-B55 about 3000-fold. Although these data do not preclude the existence of interactions outside of the active site and involving regions of Endos distant from the Gwl-phosphorylated site, it is clear that most of the binding is dictated by insertion of the phosphorylated residue into the active site, where it can then be dephosphorylated. Indicative of the strong affinity between activated Endos and PP2A-B55, the IC$_{50}$ of thiophosphorylated Endos is only about twice that of okadaic acid measured in similar assays (107 ± 10 pM *Figure 8A*), in close accordance with previous determinations (*Cohen et al., 1989*; *Kam et al., 1993*; *Walsh et al., 1997*).

These results suggest that Gwl-phosphorylated pEndos inhibits the dephosphorylation of CDK-phosphorylated substrates through competition for PP2A-B55's active site (*Figure 9*). We call our model for this mechanism 'inhibition by unfair competition' to reflect the large disparities between the kinetic parameters for pEndos in comparison with other substrates. pEndos binds rapidly to all available PP2A-B55, while the release of Endos from the enzyme (either through dissociation or

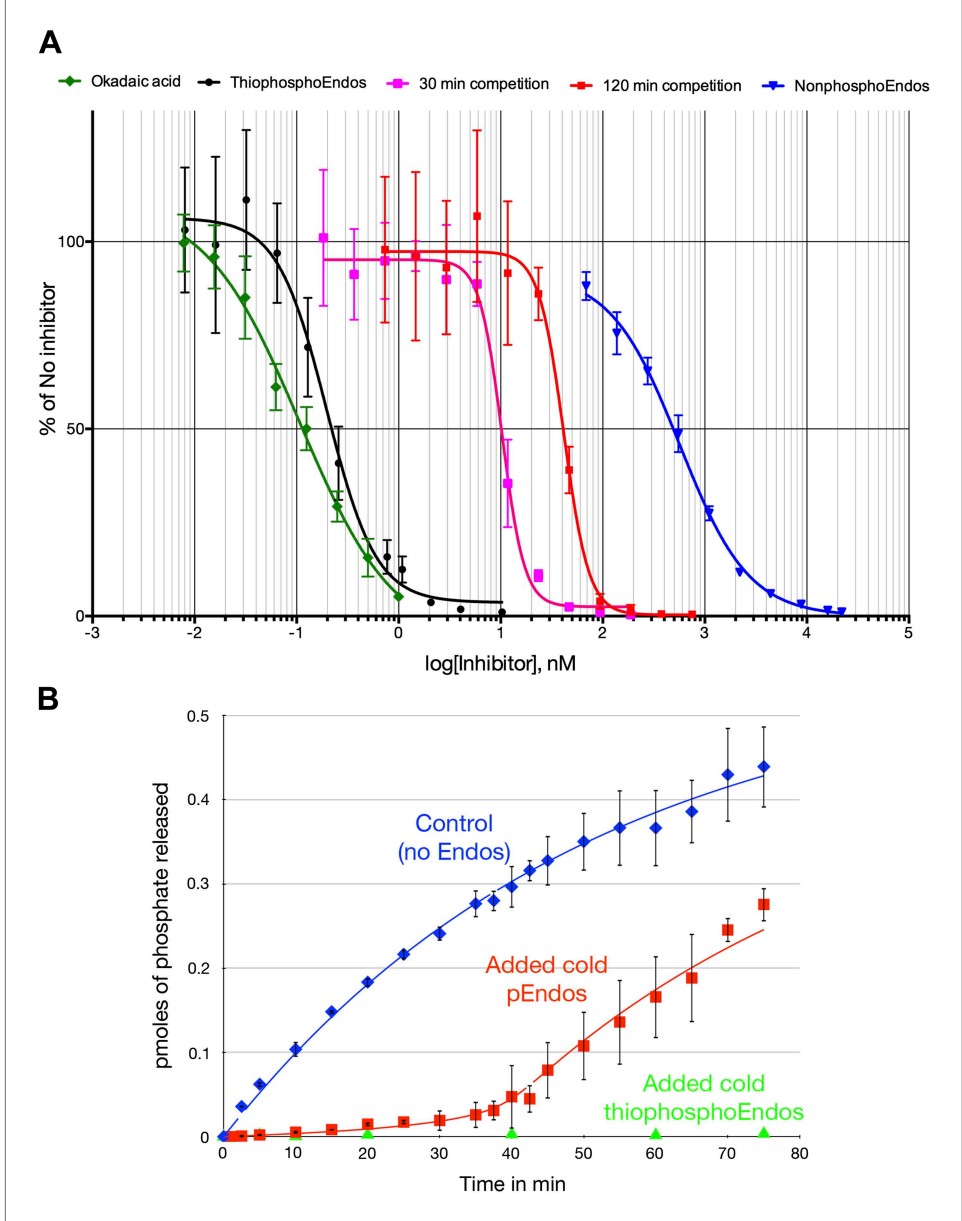

**Figure 8**. Dose-response curves for PP2A-B55 inhibitors. (**A**) The ability of PP2A-B55 (at 0.125 nM) to dephosphorylate radioactive CDKS substrate (at 5 μM) was measured in the presence of varying amounts of the following inhibitors: okadaic acid (green), thiophosphorylated Endos (black), non-radioactive pEndos (during a 30-min incubation [pink] and a 120-min incubation [red]), and unphospohorylated Endos (blue). Thiophosphorylation by Gwl kinase makes Endos into a much stronger inhibitor of PP2A-B55 than unphosphorylated Endos; the affinity of thiophosphorylated Endos is only about twofold lower than that of okadaic acid. Non-radioactive pEndos is a less efficient inhibitor of PP2A-B55 than is thiophosphorylated Endos because the enzyme dephosphorylates pEndos during the course of incubation; the longer pEndos is exposed to the enzyme, the higher is the concentration of pEndos required to achieve the same degree of inhibition. For all points shown, $n = 3$ technical replicates. (**B**) Direct demonstration of the automatic reset mechanism that allows PP2A-B55 reactivation upon anaphase onset. PP2A-B55 at 0.25 nM was mixed together on ice with buffer in the absence of Endos (control) or in the presence of 16 nM unlabeled Gwl-phosphorylated Endos or Gwl-thiophosphorylated Endos and incubated on ice for 5 min; radioactive pCDKS substrate was then added to a final concentration of 0.47 μM; and the reaction was then transferred to 30°C at $t = 0$ and aliquots assayed for the release of $^{32}$P from the pCDKS substrate. Each point represents the average of three technical replicates ($n = 3$) with standard deviations shown. The dephosphorylation of pCDKS is suppressed until the majority of pEndos is inactivated by PP2A-B55-mediated dephosphorylation.
*Figure 8. Continued on next page*

*Figure 8. Continued*

(The gradual decrease in the slopes of the control and pEndos-added curves is probably due to instability and/or degradation of the PP2A-B55 during the extended time-course.) The control and cold pEndos data were fitted to curves calculated as described in 'Materials and methods'; the best fit gave $K_m$ = 0.47 ± 0.14 nM and $k_{cat}$ = 0.03 s$^{-1}$. The stronger inhibition caused by the thiophosphorylated pEndos (green triangles) is expected because its $K_d$ (~0.12 nM) is lower than the $K_m$ of pEndos.

the removal of dephosphorylated Endos product) is very slow. Sequestration by pEndos would thus severely compromise the ability of PP2A-B55 to target alternative substrates with much higher $K_m$ values (in the 1 µM range or above).

## PP2A-B55 can reactivate itself by dephosphorylating pEndos

One prediction of the inhibition by unfair competition hypothesis is that PP2A-B55 should be able to auto-reactivate at the conclusion of M phase. By dephosphorylating pEndos inhibitor that can no longer be replenished once Gwl is inactivated, PP2A-B55 is now freed to recognize its normal, CDK-phosphorylated substrates. We tested this idea by setting up an in vitro analog of this aspect of M phase exit. We added varying amounts of non-radioactive pEndos to PP2A-B55, and monitored the

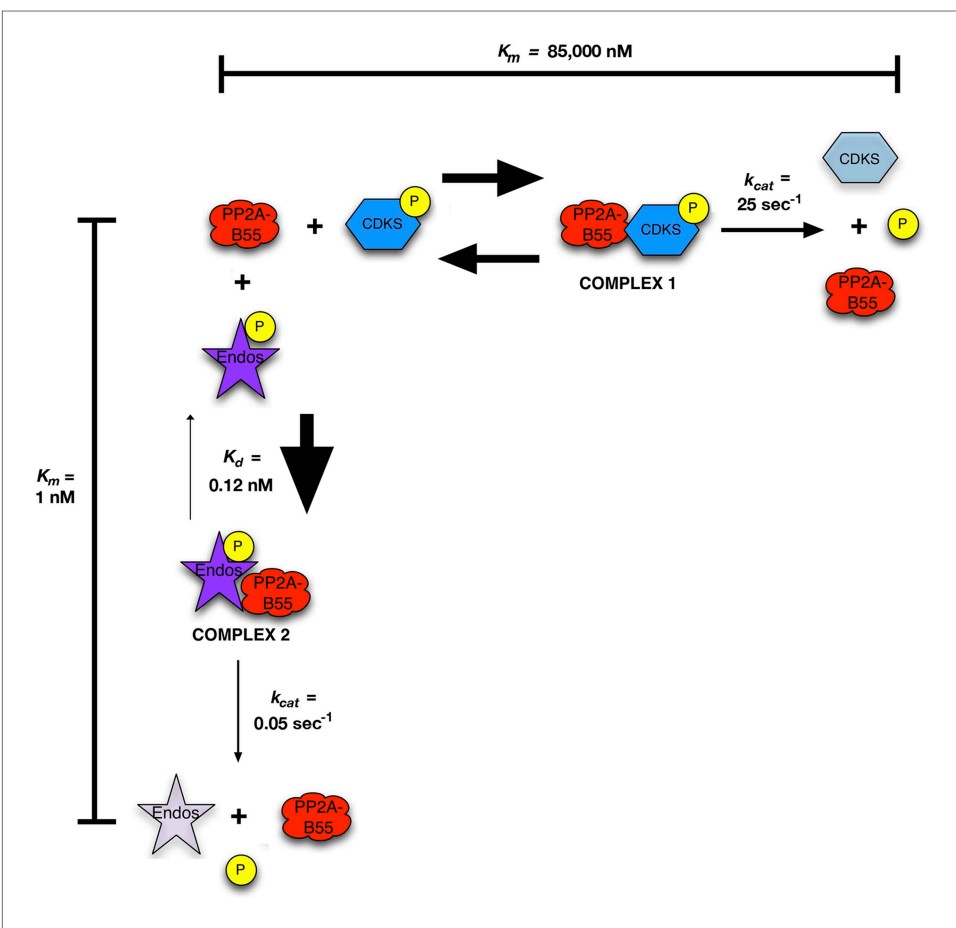

**Figure 9**. Inhibition by unfair competition. In this model, pEndos and CDK-phosphorylated substrates compete for the active site of PP2A-B55. During M phase, pEndos is in stoichiometric excess of PP2A-B55, so due to the disparities in the kinetic constants of these substrates, almost all of the enzyme will accumulate as a tight-binding complex with pEndos. Although PP2A-B55 can slowly dephosphorylate pEndos, the inhibitor is replenished by action of Gwl until this kinase is inactivated at anaphase onset (not shown).

enzyme's ability to dephosphorylate radioactive pCDKS added at the same time. The results, shown in *Figure 8A*, show clearly that pEndos indeed inhibits PP2A-B55-mediated pCDKS dephosphorylation. Much higher concentrations of pEndos than thiophosphorylated Endos are needed to achieve the same degree of inhibition, which is to be expected if pEndos is being inactivated during the incubation. As the time of incubation increases, PP2A-B55 consumes more pEndos, so even higher concentrations of pEndos are required for the same amount of inhibition. Comparing the effects of pEndos inhibition after 30- and 120-min incubations demonstrates that PP2A-B55 cannot recognize the pCDKS substrate until the majority of the pEndos in the same tube has been dephosphorylated. We estimate that the $k_{cat}$ for the dephosphorylation of pEndos in these reactions is between 0.03 and 0.12 s$^{-1}$, consistent with the range previously determined in *Table 1*.

*Figure 8B* provides straightforward evidence for the automatic reset of PP2A-B55 activity that we postulate occurs after the enzyme has succeeded in inactivating pEndos. At the beginning of this experiment, purified PP2A-B55 was mixed on ice with either buffer, nonradioactive pEndos, or nonradioactive thiophosphorylated Endos; then radioactive pCDKS substrate was added and the samples were incubated at 30°C starting at $t = 0$. The time courses of pCDKS dephosphorylation, assayed by $^{32}$P release, were then measured. A clear delay in pCDKS dephosphorylation of roughly 35 min is visible in the sample containing pEndos, after which the rate of the reaction approaches the initial rate observed in the absence of Endos inhibitors. (This delay is longer than would occur in cells because the ratio of pEndos to the enzyme is much higher than the physiological condition. Also, even stronger in vivo inhibition is predicted because the physiological pEndos concentration is many times higher than the concentration used here, which was limited by experimental constraints.) The best-fit values to these data, $K_m = 0.47 \pm 0.14$ nM and $k_{cat} = 0.03$ s$^{-1}$, are consistent with those determined by the okadaic acid competition experiments (*Table 1*; 'Materials and methods' for the calculations underlying the theoretical curves).

## Discussion

### PP2A-B55 is the predominant phosphatase targeting the Gwl phosphosite in Endos

Three independent lines of evidence indicate that the major activity contributing to the dephosphorylation of Gwl-phosphorylated Endos is a phosphatase that includes the three subunits of the PP2A-B55 heterotrimer. (1) *Inhibitor specificities*: all of the anti-Endos activity (whether in M phase or interphase) is sensitive to okadaic acid and calyculin A (*Figure 2B*, *Figure 2—figure supplement 1*), and the majority is highly sensitive to fostriecin (*Figure 2C*, *Figure 2—figure supplement 2*), suggesting that the catalytic subunit of the anti-Endos phosphatase is PP2A or its less abundant relatives PP4 or PP6. Furthermore, the facts that tautomycetin and the S68D *Drosophila* phosphomimetic Endos protein are such weak inhibitors of anti-Endos (*Figure 2*, *Figure 6*) can be most easily explained if the anti-Endos phosphatase has a very low $K_m$ for pEndos—below that for either inhibitor—as is the case for PP2A-B55. (2) *Ablation of phosphatase activities*: depletion of PP2A-A or PP2A-C from S2 tissue culture cells removes most anti-Endos from extracts (*Figure 3*), while depletion of PP4 from HeLa cells does not affect this activity (*Figure 3—figure supplement 1*). *Drosophila* larvae mutant for *twins*, which encodes the sole B55-type regulatory subunit of PP2A in flies, exhibit very little anti-Endos, while mutations in genes for other PPP family catalytic and regulatory subunits do not impede pEndos dephosphorylation in larval extracts (*Figure 4*). (3) *Biochemical purification of the activity*: anti-Endos copurifies with the anti-CDKS activity previously ascribed to PP2A-B55 heterotrimer (*Mochida and Hunt, 2007*; *Castilho et al., 2009*), while on the same column it resolves away from other PPP phosphatase subunits (*Figure 5*). Furthermore, highly purified PP2A-B55 heterotrimer displays robust anti-Endos activity whose properties match that of the predominant anti-Endos activity in extracts (*Figure 6*).

Because most of our experiments in *Figure 2* were performed with extracts prepared from unsynchronous cells (the large majority of which are in interphase), it is conceivable that a phosphatase other than PP2A-B55, that is activated only for a very short period following anaphase onset, is the enzyme responsible for dephosphorylating pEndos during M phase exit. We believe that this possibility is extremely remote for several reasons. (i) As discussed below, the inhibition by unfair competition mechanism we propose is sufficiently fast to account for the observed rapidity of M phase exit. (ii) The characteristics of the M phase and interphase anti-Endos activities measured in *Xenopus* egg extracts are very similar (*Figure 2—figure supplements 1C and 2A*); PP2A-B55's ability to dephosphorylate

pEndos is therefore constitutive with respect to the cell cycle (see also *Figure 2A*). (iii) Because of its extremely low $K_m$, pEndos is so tightly bound to PP2A-B55 during M phase that no other hypothetical phosphatase would be able to inactivate it in a reasonable time frame; the theoretical simulation in *Figure 10* below illustrates this point. We thus see no reason to postulate the existence of such a transiently activated phosphatase, and we think this possibility is incompatible with the observed relationship between PP2A-B55 and pEndos.

Some extracts contain a secondary activity (~10 to 30% of the total) that also targets the Gwl site in Endos. We have not characterized this minor activity in detail, but some evidence suggests that it may be a form of PP1: it is relatively resistant to fostriecin (*Figure 2—figure supplement 2*), and RNAi depletion of PP1-87B, the most abundant form of the PP1 catalytic subunit, removes ~20 to 25% of anti-Endos activity from *Drosophila* S2 cell extracts (*Figure 3*).

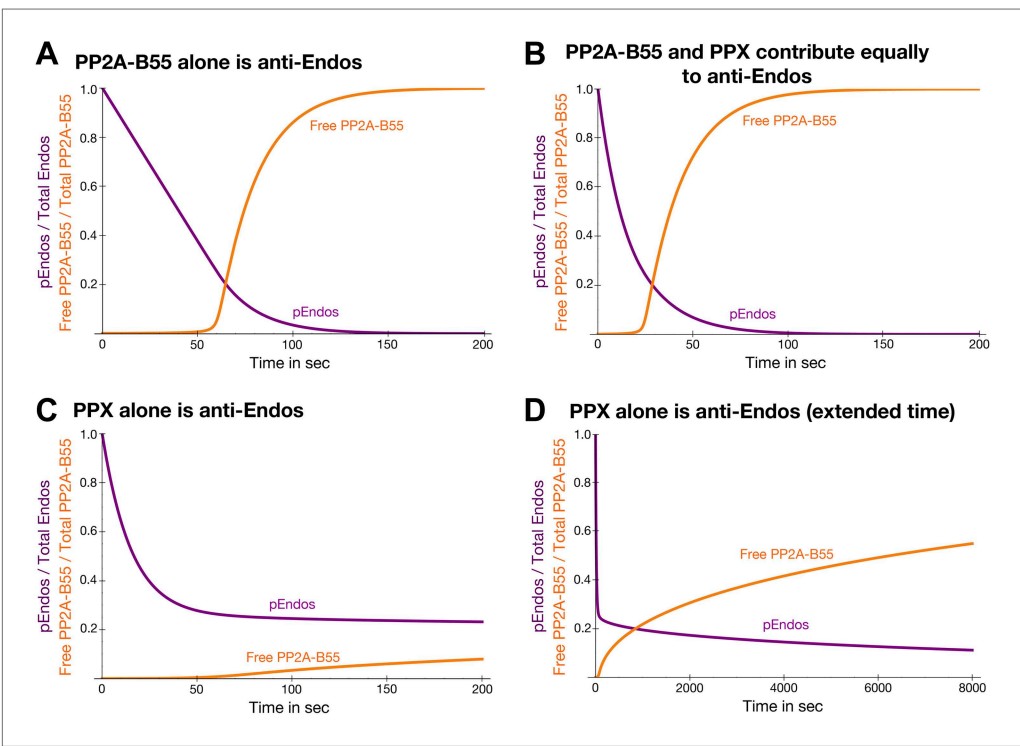

**Figure 10**. Theoretical time-course of pEndos dephosphorylation and PP2A/B55 activation at the end of M-phase. These calculations make three assumptions: First, consistent with published data (*Gharbi-Ayachi et al., 2010*; *Mochida et al., 2010*), Endos was completely phosphorylated during M phase. Second, because the $K_m$ of PP2A-B55 for pEndos (~1 nM, *Table 1*) is much smaller than the pEndos concentration (1 μM), essentially all the PP2A-B55 was bound to pEndos during M phase. Third, the Gwl phosphorylation of Endos stopped at the end of M-phase ($t = 0$) (*Castilho et al., 2009*; *Mochida et al., 2009*; *Gharbi-Ayachi et al., 2010*; *Mochida et al., 2010*). The curves show the fraction of Endos that remains phosphorylated (purple) and the fraction of PP2A-B55 that is released from pEndos sequestration (orange) as a function of time $t$ in sec. (**A**) Scenario: PP2A-B55 is the only enzyme that can dephosphorylate pEndos. Parameter values estimated from the experimental data were used: Total PP2A/B55 concentration, 250 nM; initial total pEndos concentration, 1 μM; $K_m$, 1 nM; and $k_{cat}$, 0.05 s$^{-1}$. (**B**) Scenario: pEndos is a target of both PP2A-B55 and equal amounts of PPX, a second phosphatase with parameters $K_m$ = 85 μM and $k_{cat}$ = 22.5 s$^{-1}$ (based on the dephosphorylation of pCDKS by PP2A-B55 in *Table 1*). (**C** and **D**) Scenario: pEndos is only an inhibitor and not a substrate of PP2A-B55, and only PPX dephosphorylates pEndos. In this case, the $K_d$ = 0.12 nM of thiophosphorylated Endos was used; this value was calculated from the IC$_{50}$ for thiophosphorylated Endos from the dose-response curve in *Figure 7* according to the equation described in *Sasaki et al. (1994)* and *Takai et al. (1995)*. The time frame in **D** is an extension of that in **C**, showing that the time required for desequestration of 50% of PP2A-B55 activity would be ~6200 s under this last scenario.

The following figure supplements are available for figure 10:

**Figure supplement 1**. PP2A-B55 sequesters pEndos from the action of other phosphatases.

Regardless of its exact composition, we believe this secondary activity is not very important to the ultimate goal of regulating PP2A-B55 function. *Figure 10—figure supplement 1* models the scenario that another phosphatase (called PPX in the figure, but presumptively PP1) exists that can target pEndos. We chose the $K_m$ and $k_{cat}$ values for this putative reaction to match those for the dephosphorylation of pCDKS by PP2A-B55 (from *Table 1*), and the concentration of PPX to be the same as that of PP2A-B55 (about 100–250 nM by our estimation, depending on cell type), but in fact the basic conclusions would be essentially unaltered by 10-fold changes in any of these assumptions. At high pEndos concentrations above the intracellular concentration of PP2A-B55, PPX could dephosphorylate the excess pEndos that was not bound to PP2A-B55. However, this dephosphorylation would have little regulatory consequence: as soon as the pEndos concentration is reduced to the intracellular concentration of PP2A-B55, the remaining pEndos would be protected from dephosphorylation by PPX due to the very tight (nanomolar or subnanomolar) binding of pEndos to PP2A-B55. The consequences of this conclusion to the dynamics of the M phase-to-interphase transition will be further explored later in the 'Discussion'.

Since the original submission of this manuscript, three other papers have been published whose results bear on our identification of PP2A-B55 as the major anti-Endos phosphatase. The results from two of these studies support this conclusion. First, S Mochida measured the $IC_{50}$ of pEndos and thiophosphorylated Endos and obtained values very close to those shown in our *Figure 8A* (*Mochida, 2013*). Moreover, he demonstrated that Endos bound by PP2A-B55 is in close proximity to both the B55 and C (catalytic) subunits of the enzyme, consistent with a location at the active site. An extremely tight association of phosphorylated Endos with the PP2A-B55 active site is an essential feature of our inhibition by unfair competition model. Second, during the course of an investigation that will be discussed further below, the laboratory of F A Barr provisionally identified PP2A-B55 as the phosphatase responsible for dephosphorylating pEndos in a mammalian cell system (*Cundell et al., 2013*).

## Fcp1 is not a major anti-Endos phosphatase

A third recent publication (*Hegarat et al., 2014*) came to a conclusion incompatible with that presented here: these authors maintain that the anti-Endos enzyme is Fcp1, a phosphatase known to target phosphorylations of the C-terminal domain (CTD) of RNA polymerase II. We believe their report is incorrect for several reasons. (i) We show here strong evidence that PP2A-B55 is responsible for the anti-Endos activity. Furthermore, *Figure 10C,D* demonstrates that the very tight binding of pEndos to the PP2A-B55 active site blocks its binding to other phosphatases. No other normal-binding phosphatase could contribute to pEndos inactivation rapidly enough to account for the timing of M phase exit. (ii) Among other issues, this new publication is internally inconsistent. The authors show that the anti-Endos activity they ascribe to Fcp1 is okadaic acid sensitive (supplemental Figure S3 in reference *Hegarat et al., 2014*), yet the literature contains many reports that Fcp1 is completely resistant to okadaic acid (e.g., *Palancade et al., 2001*; *Washington et al., 2002*; *Kong et al., 2005*; *Visconti et al., 2012*), a fact that we verify in our *Figure 11E*.

We have also obtained results that more directly exclude Fcp1 as a major anti-Endos phosphatase. (iii) In mono-Q chromatography, Fcp1 does not co-fractionate with the pEndos-dephosphorylating activity. *Figure 5* shows that fractions with peak anti-Endos activity (such as fractions from 37 to 39) have no detectable Fcp1 protein. (iv) RNAi depletion of more than 90% of the Fcp1 in *Drosophila* S2 cells has no obvious effect on the ability of extracts made from these cells to dephosphorylate pEndos (*Figure 11A,B*). (v) Purified *S. pombe* Fcp1 enzyme has detectable anti-Endos activity, but this is very low. At a physiological concentration of the pEndos (1 μM), each molecule of Fcp1 has less than 1/1,000[th] the efficiency of a molecule of PP2A-B55 in dephosphorylating this substrate (*Figure 11C,D*). Moreover, cells contain more than 10 times more PP2A-B55 than Fcp1 molecules (based on values in the Pax-DB database of protein abundances determined by mass spectrometry; see www.pax-db.org and reference *Wang et al., 2012*). By this estimate, less than 0.01% of the total anti-Endos activity in cells can be ascribed to Fcp1. It should be cautioned that our measurements of Fcp1 enzyme kinetics were made using the heterologous *S. pombe* enzyme, but the structure of Fcp1 is well conserved in evolution.

## A simple competition mechanism explains how pEndos acts both as an inhibitor and a substrate for PP2A-B55

In *Figure 9*, we propose a straightforward model for the relationship between pEndos and PP2A-B55. In essence, pEndos competes with a large class of CDK-catalyzed mitotic phosphosites (of which

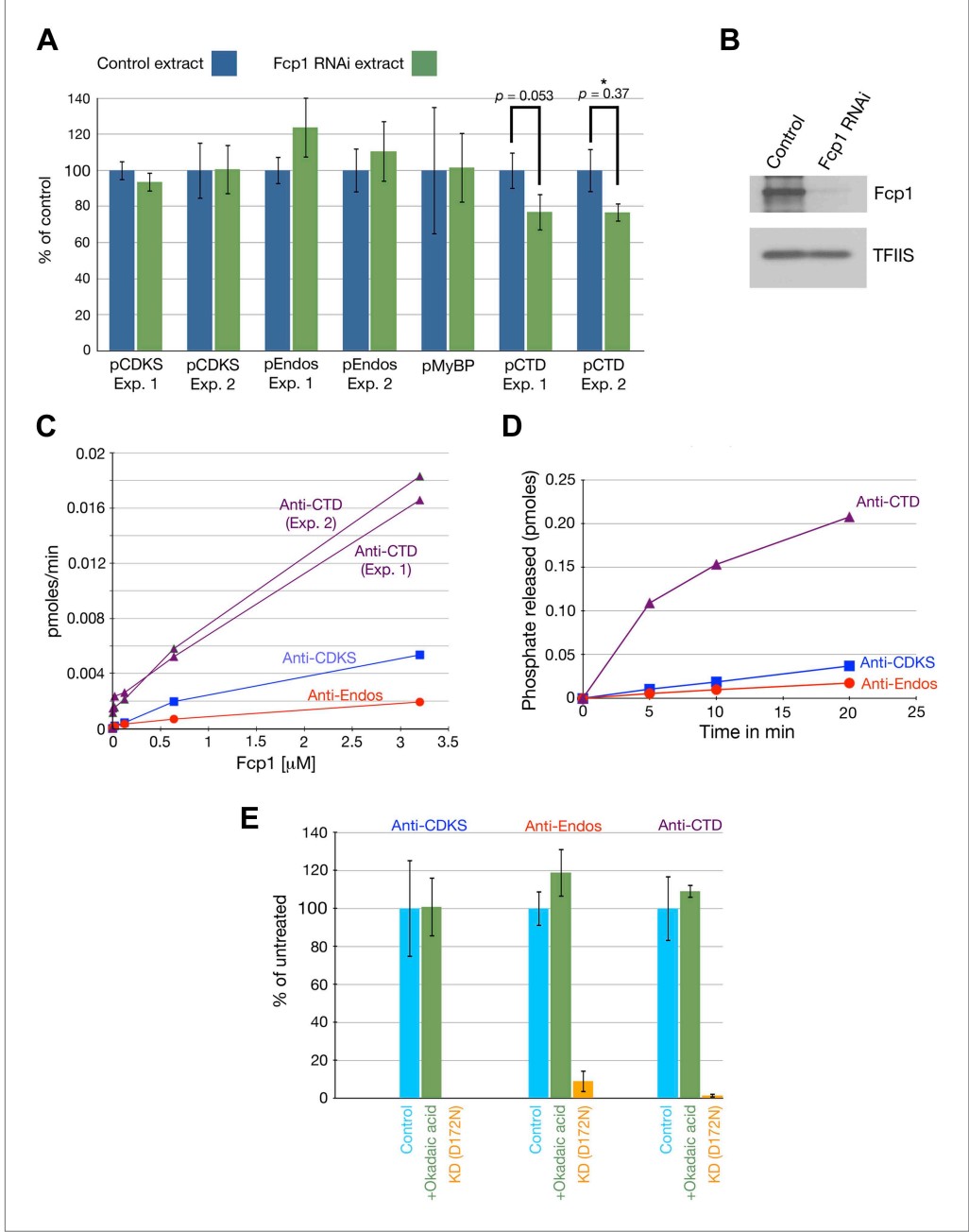

**Figure 11**. Fcp1 is not the anti-Endos phosphatase. (**A** and **B**). Depletion of Fcp1 does not decrease anti-Endos activity. (**A**) Control extracts (made from S2 cells treated with a mock double-stranded RNA) or Fcp1 RNAi extracts (made from S2 cells treated with double-stranded RNA corresponding to the *Drosophila Fcp1* gene [*CG12252*] were assayed for phosphatase activities using pCDKS, pEndos, and pCTD (the C-terminal domain of RNA polymerase II phosphorylated in vitro by mitogen-activated kinase 2 [MAPK2]) substrates. No significant differences were observed in terms of anti-CDKS or anti-Endos activities. Fcp1 depletion slightly decreases activity against the CTD substrate, consistent with the fact that Fcp1 is only one out of several known phosphatases known to target the CTD (**Hsin and Manley, 2012**). Each bar represents the average of three technical replicates with the standard deviation shown. (**B**) Western blot of the control and Fcp1 RNAi extracts assayed in part **A**, showing that more than 90% of the Fcp1 was removed by the treatment with *Fcp1* double-stranded RNA. The loading control shows the signal revealed by antibody against the RNA polymerase II elongation factor TFIIS. (**C** and **D**). Purified *S. pombe* Fcp1 enzyme has minimal anti-Endos activity. Activities of purified Fcp1 against labeled pEndos, pCDKS, and pCTD. Each point represents a single assay. Note that the concentrations of Fcp1 employed in these experiments

*Figure 11. Continued on next page*

*Figure 11. Continued*

are in the micromolar range, as opposed to the subnanomolar-to-nanomolar concentrations of purified PP2A-B55 used in other figures. Consistent with previous reports for this enzyme (*Hausmann and Shuman, 2002*), the specific activity of Fcp1 against the CTD substrate is low (but much higher than that against pEndos). (**E**) The activities of purified wild-type Fcp1 against all three substrates are, as expected, resistant to high concentrations of okadaic acid (10 μM). A preparation of kinase-dead (KD) *S. pombe* Fcp1 with the D172N mutation (*Suh et al., 2005*) made in parallel has no activity against any of these three substrates.

pCDKS and pSer50 Fizzy are examples) for access to the phosphatase active site. pEndos can successfully compete for the site because its affinity for PP2A-B55 is much higher than those of the competing substrates; that is, the $K_m$ for pEndos is extremely low. However, PP2A-B55 dephosphorylates pEndos at a much slower rate (low $k_{cat}$) than the CDK phosphosites, so the tight interaction of PP2A-B55 and pEndos would be prolonged. Thus, pEndos would soon sequester the phosphatase from competing substrates, protecting such substrates against premature dephosphorylation. pEndos in this way permits M phase to begin and to be maintained as long as Gwl kinase is active. We call this mechanism 'inhibition by unfair competition'. In its essence, the model shown in *Figure 9* clarifies how PP2A-B55 can be 'off' during M phase for CDK-phosphorylated substrates, but 'on' at the same time for the Gwl-phosphorylated pEndos substrate.

In the context of the cell, the inhibition by unfair competition mechanism requires that pEndos be present in molar excess over PP2A-B55 in mitosis to account for the near-total absence of anti-CDKS activity observed during M phase (*Burgess et al., 2010*; *Mochida and Hunt, 2007*; *Mochida et al., 2009*; *Figure 2A*). Considerable evidence exists that this requirement is met in many cell types. From our own quantitation on Western blots, we estimate the intracellular concentration of the PP2A B55 subunit to be between 100 and 250 nM and that of Endos to be between 500 nM and 1 μM, depending on the cell type. (One example of this analysis is seen in *Figure 2—figure supplement 6*). The 5:1 ratio of Endos to PP2A-B55 observed in that figure is roughly consistent with estimates from other laboratories (*Cundell et al., 2013*) and with estimates from mass spectrometry studies of whole cell extracts, after accounting for all Endos and PP2A-B55 family members (*Brunner et al., 2007*; *Beck et al., 2011*; *Kolker et al., 2012*; *Wang et al., 2012*). For example, the Pax-DB database integrating the results of many mass spectrometry experiments on a variety of tissue types estimates that the abundance of Endos in *Drosophila* cells is 295 ppm while that of Twins (B55) is 123 ppm; for human cells, the integrated datasets yield values of 83.2 ppm for Endos-family proteins (ENSA and ARPP-19) and 23.1 ppm for B55-family proteins (www.pax-db.org; *Wang et al., 2012*). Although we ourselves have not determined the fraction of the total Endos protein that is phosphorylated during M phase, other investigators have found that this proportion is roughly 50% in extracts of synchronized mammalian tissue culture cells (perhaps some of which may not have been in M phase) (*Cundell et al., 2013*), and approaches 100% in frog egg M phase extracts (*Mochida et al., 2010*). Sufficient 'headroom' thus exists to conclude that the molar concentration of pEndos during M phase in fact exceeds that of the PP2A-B55 phosphatase.

A major virtue of the inhibition by unfair competition model is that it explains not only how pEndos inhibits PP2A-B55 dephosphorylation of pCDKS-class substrates, but also how pEndos can itself become inactivated: the system has an automatic reset that is intrinsic to the mechanism that inactivates PP2A-B55 phosphatase in the first place. When Gwl is inactivated at anaphase onset, the pEndos dephosphorylated by PP2A-B55 can no longer be replaced. PP2A-B55 is now free to work on CDK-phosphorylated substrates and thus to promote the M phase-to-interphase transition. The experiments presented in *Figure 8*, in which we added non-radioactive pEndos and radioactive pCDKS to the same tube of PP2A-B55, provide a direct in vitro test of the proposed mechanism by mimicking the events that occur during M phase exit. The results show that PP2A-B55 targets pCDKS only after the enzyme has dephosphorylated almost all of the pEndos.

## Rapid M phase exit requires the inhibition by unfair competition mechanism

M phase exit is surprisingly rapid given the large number of phosphorylations which must be reversed; in *Xenopus* cycling extracts, for example, the M phase-to-interphase transition is completed within less

than 5 min (*Zhao et al., 2008*; *Mochida et al., 2009*). Even though the turnover rate of pEndos dephosphorylation by PP2A-B55 is quite slow (~0.05 s$^{-1}$), the inhibition by unfair competition mechanism is nevertheless compatible with the rapidity of M phase exit. The reason is that Endos is present in cells only in at most a fivefold stoichiometric excess with respect to the phosphatase, as was just discussed. Each molecule of PP2A-B55 thus needs to dephosphorylate only a few molecules of pEndos to effect the M phase-to-interphase transition.

To explore this idea quantitatively, we modeled the dynamics of a simplified system consisting solely of Endos and PP2A-B55 at estimated physiological conditions (*Figure 10A*). Because the $K_m$ of PP2A-B55 for pEndos is four-to-five orders of magnitude smaller than the $K_m$ for other substrates, typified by pCDKS (*Table 1*), and the pEndos concentration during M phase is in excess of the phosphatase concentration, we suggest that the approximation of ignoring the binding of PP2A-B55 to other substrates is appropriate, and that this simplified model should capture the essence of the regulatory process. The calculation begins at the end of M-phase ($t = 0$) with the inactivation of Gwl. Almost all the PP2A-B55 is sequestered by pEndos until PP2A-B55-catalyzed dephosphorylation causes the pEndos concentration to decrease from 1 µM to ~250 nM, the intracellular PP2A-B55 concentration. Beyond this point, PP2A-B55 is rapidly released from sequestration; half is available to act on other substrates within 74 s (*Figure 10A*). Note that this calculation essentially recapitulates the actual experiments shown in *Figure 8*, but here the inhibition of PP22A-B55 is stronger and the time to release is faster because physiological, not laboratory, conditions are being modeled.

We previously showed in *Figure 2—figure supplement 2* that cells likely harbor a pEndos-targeting phosphatase other than PP2A-B55; because this secondary activity is fostriecin-resistant, we speculate that it may be a form of PP1. *Figure 10B* shows how the time courses of pEndos dephosphorylation and PP2A-B55 desequestration are modified if this second phosphatase (again called PPX) is present at the same concentration as PP2A-B55 and acts on pEndos with kinetic parameters matching the activity of PP2A-B55 on pCDKS. In this case, the lag in PP2A-B55 desequestration shown in *Figure 10B* is shortened, because PPX can dephosphorylate the pEndos that is not bound to PP2A/B55. However, as illustrated in *Figure 10—figure supplement 1*, the presence of PPX makes very little difference once the pEndos concentration decreases to the PP2A-B55 concentration of 250 nM, since almost all the remaining pEndos is bound to PP2A-B55. In this scenario, half of the PPA-B55 is released from pEndos by 38 s, producing a modest acceleration in M phase exit. We regard the situation shown in *Figure 10B* as a reasonable description of the actual events in the cell, as it accounts for contributions from both PP2A-B55 and other phosphatases in pEndos inactivation.

Further dynamic modeling surprisingly revealed a key insight: cells would be unable to exit M phase in a timely manner unless pEndos was in fact dephosphorylated by PP2A-B55 as demanded by unfair competition. *Figure 10C,D* assumes that pEndos is only an inhibitor and not a substrate of PP2A-B55; thus pEndos binds and sequesters PP2A-B55, but is only dephosphorylated by PPX (whose concentrations and properties are identical to those in *Figure 10B*). In this scenario, PP2A-B55 would protect the bound pEndos from dephosphorylation, so many hours—not just a few minutes—would be required for PP2A-B55 desequestration and M phase exit. In other words, the dephosphorylation of pEndos by PP2A-B55 is an absolute requirement for rapid completion of the M phase-to-interphase transition.

Our identification of PP2A-B55 as the anti-Endos phosphatase poses a dilemma in terms of M phase exit: the inhibition by unfair competition model describes events that must occur downstream of the inactivation of Gwl kinase, but what then can inactivate Gwl upon anaphase onset? According to our model, the rate-limiting Gwl-inactivating phosphatase cannot be PP2A-B55, because the system would be futile. Gwl inactivation could not proceed until pEndos was depleted, but pEndos cannot be inactivated as long as Gwl is active. An argument can be made that PP2A-B55 should be the Gwl-inactivating phosphatase because Gwl is activated by CDKs (*Blake-Hodek et al., 2012*), and PP2A-B55 targets at least a subset of CDK phosphosites (*Mochida and Hunt, 2007*; *Castilho et al., 2009*; *Mochida et al., 2009*). Indeed, one group has recently reported some evidence in favor of this idea (*Hegarat et al., 2014*). However, it should be emphasized that PP2A-B55 does not act on all, or even perhaps the majority, of CDK phosphosites, as recently most forcefully demonstrated by *Cundell et al. (2013)*. The obvious resolution of this dilemma is therefore that a phosphatase other than PP2A-B55 inactivates Gwl during M phase exit. We have obtained preliminary evidence in support of this idea in two ways (data not shown). First, in our hands purified PP2A-B55 in vitro is very inefficient in inactivating Gwl. Second, when active Gwl is added to interphase extracts from a variety of cell types, it becomes rapidly inactivated and dephosphorylated in a process that is

insensitive to okadaic acid (and cannot therefore be controlled by PP2A-B55). Some evidence in favor of the existence of an okadaic acid-resistant anti-Gwl phosphatase has also recently been obtained by another group (*Hegarat et al., 2014*); the identity of this Gwl-inactivating enzyme is currently unknown.

F A Barr et al. have recently proposed an elegant model in which the events of anaphase onset are ordered in time, with later events such as cytokinesis being controlled by the Gwl-Endos pathway (*Cundell et al., 2013*). Our conclusions are in essence consistent with their hypothesis, and the automatic reset mechanism we propose provides an explanation for part of the delay they observe between CDK1-Cyclin B inactivation and the activation of PP2A-B55, and thus for the dephosphorylation of late substrates involved in cytokinesis such as PRC1. That is, the major determining factors for this delay would be the rate of Endos dephosphorylation by PP2A-B55 described here coupled with the time required to inactivate Gwl through mechanisms we do not yet understand.

## Other AGC kinases may indirectly inhibit other PPP phosphatases through similar mechanisms

The virtues of inhibition by unfair competition are sufficiently compelling that one might ask whether other phosphatases are controlled in the same fashion. We believe this supposition is likely, based on findings published 30 years ago concerning inhibition of PP1 by a polypeptide called Inhibitor-1 that is, like Endosulfine, small and heat-stable (*Foulkes et al., 1983*). Inhibition occurs only after Inhibitor-1 is phosphorylated at a conserved site by the cAMP-dependent protein kinase (PKA), an enzyme whose structure and activation mechanism are closely related to those of its fellow AGC kinase family member, Gwl (*Blake-Hodek et al., 2012*). The investigators found that purified PP1 could dephosphorylate PKA-phosphorylated Inhibitor-1, and the values of $K_m$ and $k_{cat}$ were both considerably lower for this reaction than for dephosphorylation of the more normal substrate phosphorylase *a* (*Foulkes et al., 1983*). In short, just as we report here for pEndos and PP2A-B55, they concluded that pInhibitor-1 acts both as an inhibitor and a substrate for PP1.

These observations have long since been disregarded because PP1's activity in inactivating pInhibitor-1 was very slow, and other phosphatases (PP2A and calcineurin) had substantial pInhibitor-1 dephosphorylating activity (*Ingebritsen et al., 1983*; *Cohen, 1989*). In light of the arguments summarized in *Figure 10* that a phosphatase will effectively sequester a tightly binding inhibitor from other phosphatases, we believe the biological significance of PP1's activity in dephosphorylating pInhibitor-1 should be re-evaluated. Of interest, recent studies in *Xenopus* egg extracts have indicated that pInhibitor-1 dephosphorylation at the end of M phase is in fact dependent upon activity of PP1, although this work could not distinguish whether the effect was direct or indirect (*Wu et al., 2009*). The inhibition by unfair competition mechanism may well prove to have been utilized by AGC kinases other than Gwl to control phosphatases other than PP2A-B55.

# Materials and methods

## Fly stocks

Extracts from third instar larvae of the following genotypes were assayed for phosphatase activities: PP2A B55 regulatory subunit (*twins*): $tws^P/tws^{196}$ (*Uemura et al., 1993*; *Kim et al., 2012*). PP2A B56 regulatory subunit type 1 (*widerborst*): $wdb^{12–1}/wdb^{12–1}$ (*Kotadia et al., 2008*). PP2A B56 regulatory subunit type 2 (*well-rounded*): $wrd^{KG01108}/wrd^{KG01108}$ (*Rollmann et al., 2007*). PP2A B''' (striatin) regulatory subunit (*connector of kinase to AP-1*): $cka^{05836}/cka^{S1883}$ (*Perrimon et al., 1996*; *Spradling et al., 1999*). PP4 regulatory subunit PP4R3 (*falafel*): $flfl^{N42}/flfl^{795}$ (*Sousa-Nunes et al., 2009*). PP1 catalytic subunit (*protein phosphatase 1 at 87B*): $PP1-87B^1/PP1-87B^{87Bg-3}$ (*Gausz et al., 1981*; *Reuter et al., 1987*). PP1 catalytic subunit (protein phosphatase 1α at 96A): $PP1-96A^2/PP1-96A^2$ (*Kirchner et al., 2007a*). PP1 catalytic subunit (*flapwing*; also known as *PP1β-9C*): $flw^6/flw^{GO172}$ (*Raghavan et al., 2000*; *Bennett et al., 2003*; *Kirchner et al., 2007b*).

Stocks with these mutant alleles were obtained from the *Drosophila* Stock Center (Bloomington, IN) and rebalanced over chromosomes containing *Tubby* (*Tb*) to facilitate the identification of homozygous or *trans*-heterozygous mutant third instar larvae. For the genes on the third chromosome (*tws, wdb, flfl, PP1-87B, PP1α-96A*) the balancers used either were $TM6B$, $Tb^1$ $Antp^{Hu}$ or $TM6C$, $Tb^1$ $Sb^1$. For the *cka* gene on the second chromosome, the balancer was $CyO-TbA$, while for the *flw* gene on the X chromosome, the balancer was $FM7A-TbA$ (*Lattao et al., 2011*). The $wrd^{KG01108}$ mutation is homozygous viable. For certain genes, *trans*-heterozygous combinations of mutant alleles were employed as indicated

by the genotypes above to obtain mutant third instar larvae; this was necessary because chromosomes bearing the single mutations have over time accumulated second-site lesions causing earlier lethality.

## Cell lines

HeLa CCL-2 and HEK293 CRL-1573 cells were obtained from the American Type Culture Collection (Manassas, VA). PP5 +/+, +/− and −/− mouse MEF cells (*Yong et al., 2007*) were the kind gift of Dr Wenian Shou, (Indiana University-Purdue University, Indianapolis, IN). All mammalian cell lines were grown in Dulbecco's Modified Eagle Medium +10% Fetal Bovine Serum (Invitrogen, Carlsbad, CA). *Drosophila* S2 cells were grown in Insect-Xpress with L-Glutamine (Lonza BioWhittaker, Walkersville, MD; catalog number 12-730Q). Both mammalian and insect cells were grown in the presence of 100 units/ml of penicillin and 100 μg/ml of streptomycin and a 1:100 dilution of the antimycotic Fungizone (Gibco, Gaithersburg, MD).

## Phosphatase substrates

Substrates for phosphatase reactions were purified as previously described (*Castilho et al., 2009*) from *E. coli* transformed with recombinant plasmids constructed from the vector pMAL-C2X (New England Biolabs, Ipswich, MA) and expressing maltose binding protein fusions with the following peptides/proteins: full-length *Xenopus* Endosulfine (Endos; *Castilho et al., 2009*; *Kim et al., 2012*); the region from *Xenopus* PP1γ surrounding the phosphosite Thr311 (we term this substrate CDKS; *Mochida and Hunt, 2007*; *Castilho et al., 2009*; *Mochida et al., 2009*); the region surrounding the Ser50 phosphosite of *Xenopus* Fizzy (Fzy; *Mochida and Hunt, 2007*; *Castilho et al., 2009*; *Mochida et al., 2009*); amino acids 1–27 of *Drosophila* Histone H3; and full-length *H. sapiens* histone H1.0 (purified Histone H1.0 protein obtained from New England Biolabs yielded identical results). In a few experiments included in *Table 1*, *Xenopus* Endos was cloned into the vector pQE30 (Qiagen, Valencia, CA) so as to express hexahistidine-tagged protein. Bovine myelin basic protein (MyBP) was from Millipore (Temecula, CA). Substrates for phosphatase assays of the RNA polymerase II C-terminal domain (CTD) were purified as GST-CTD fusions as previously described (*Suh et al., 2005*).

The purified proteins were $^{32}$P-phosphorylated in vitro by Gwl kinase (Endos), by CDK1-Cyclin A (CDKS, Fzy Ser50, and Histone H1.0), by Protein Kinase A (New England Biolabs; Histone H3 and MyBP), or by mitogen activated kinase 2 (MAPK2; New England Biolabs; RNA polymerase II CTD) using methods previously described (*Suh et al., 2005*; *Castilho et al., 2009*). Proteins were purified away from the kinase and other reaction components, and desalted and concentrated using Amicon Ultracel 10K centrifugal filter columns (Millipore) and substrate buffer (20 mM Tris, pH 7.5; 150 mM NaCl). The incorporation of phosphate was determined to be between 0.5 and 1.0 per mole of protein based on measurements of protein concentration and specific activity. For some studies, the Endos fusion protein was thiophosphorylated by Gwl in the presence of 1 mM ATP-[γ]-$^{35}$S instead of ATP. Dephosphorylation of $^{35}$S-thio-phosphorylation by purified PP2A-B55 was not detectable.

## Cell-free extracts

*Xenopus* CSF and interphase extracts were prepared as described (*Castilho et al., 2009*). HeLa cells were extracted in ice-cold phosphatase buffer (50 mM Tris, pH 7.5, 150 mM NaCl, 1 mM EDTA, 0.1 mM EGTA, 0.25% NP-40) according to *Singh et al. (2010)*, or M-PER mammalian cell lysis buffer (Thermo Scientific, Rockford, IL) with the same results. A 1:100 dilution of EDTA-free Proteoblock protease inhibitor (Thermo Scientific) was added to both extraction buffers. *Drosophila* whole larvae or larval brains were homogenized with a pestle in phosphatase buffer as above, to which was added a 1:100 dilution of phenylmethanesulfonyl fluoride (PMSF; Thermo Scientific), a 1:50 dilution of Halt protease inhibitor cocktail (Thermo Scientific), and a 1:100 dilution of a 100 mg/ml stock solution of soybean trypsin inhibitor (AMRESCO, Solon, OH; catalog number M191).

## Phosphatase assays

Extracts prepared in phosphate buffer (see the section on cell-free extracts above) or purified enzymes appropriately diluted in PP2A reaction buffer (see below) were used for each phosphatase assay. A no-extract control was always included to verify that the substrate did not itself contain phosphatase activity. Unless otherwise noted, radiolabeled substrates were added to a final concentration of 5 μM in the reactions. Phosphatase assays were stopped with trichloracetic acid (TCA) after a time at

which <20% of the substrate was dephosphorylated. The supernatant was mixed with ammonium molybdate and extracted with heptane-isobutyl alcohol according to Mochida and Hunt (*Mochida and Hunt, 2007*) and the $^{32}$P measured in a scintillation counter.

In certain experiments, the following phosphatase inhibitors were used after being dissolved in dimethyl sulfoxide (DMSO): fostriecin, tautomycin, and tautomycetin (all from Santa Cruz Biochemicals, Dallas, TX), and okadaic acid (LC Labs, Boston, MA). The phosphomimetic *Drosophila* S68D Endos protein has been previously described (*Kim et al., 2012*). Inhibitors were pre-incubated on ice with extracts for 10 min prior to addition of the substrate.

## RNA interference

siRNA oligonucleotides (Thermo Scientific) for PP1, PP2A (α and β isoforms), PP4, PP5, and PP6 catalytic subunits were transfected into HeLa cells using Dharmafect following the manufacturer's protocols (Dharmacon, now Thermo Scientific) and harvested after 72 hr. The cells were lysed in phosphatase buffer and the lysates used to determine activity against $^{32}$P-labeled substrates in a phosphatase assay as described above, and for knockdown efficiency by Western blotting using the antibodies listed below.

The technique used to effect RNAi in *Drosophila* S2 tissue culture cells has been described previously (*Williams et al., 2003*). dsRNA was produced from cDNA clones for the given phosphatase subunit by PCR amplification using primers containing a T7 RNA polymerase promoter site at the 5′ end. The primer pair used for the catalytic subunit of PP2A (*microtubule star*) was:

5′ TAATACGACTCACTATAGGGAGACCTACGCAGCTTACATTTACACATA 3′ and 5′ TAATACGACT-CACTATAGGGAGATAGGTTCGATTGGATTGTATCATTT 3′.

For the A subunit of PP2A (*PP2A-29B*):

5′ TAATACGACTCACTATAGGGAGACAGAGTTTGCCATGTACTTGATTC 3′ and 5′ TAATACGACT-CACTATAGGGAGAGGAATCAAATCGGACTTCAGATACT 3′.

For the major catalytic subunit of PP1 (*PP1-87B*):

5′ TAATACGACTCACTATAGGGAGAACCACGAGCAGTCTTTTTCTATCTA 3′ and 5′ TAATACGACT-CACTATAGGGAGAGTG GCTTTTAATCATGGTATTTGTC 3′.

RNAi depletion of Fcp1 from S2 tissue culture cells has been previously described (*Fuda et al., 2012*). In all cases, S2 cell pellets were lysed and analyzed as just described for HeLa cells.

## Mono-Q chromatography

Fractionation of HeLa cell lysates by Mono-Q chromatography was performed using modifications of published protocols (*Che et al., 1998*; *Guo et al., 2002*). HeLa cells were lysed in buffer containing 50 mM Tris-HCl, pH 7.4; 100 mM NaCl; 1 mM EDTA; 0.5 mM EGTA; 0.25% NP-40; and 10% glycerol. Cell lysate was then diluted to 20 ml with Buffer A (25 mM Tris-HCl, pH 7.4; 150 mM NaCl; 1 mM DTT; and 10% glycerol). Diluted lysate was loaded onto a 5-ml HiTrap Q HP column (GE Healthcare, Uppsala, Sweden) equilibrated with Buffer A at a flow rate of 0.5 ml/min. The column was then washed for with Buffer A for 16 min (0.5 ml/min) followed by elution with a linear gradient to 500 mM NaCl over 30 min, and the column was then subjected to an additional 30 min wash at 500 mM NaCl. 0.5 ml fractions were collected every minute. The fractions were split into 100 μl aliquots, flash frozen in liquid nitrogen, and then stored at −80°C for later analysis.

## Phosphatase purification

pcDNA5 (vector alone), pcDNA5/FLAG-B55α, pcDNA5/FLAG-B55δ, and pcDNA5/FLAG-B56β plasmid DNAs were transfected into HEK293 CRL-1573 cells using Lipofectamine 2000 (Invitrogen) using the manufacturer's protocols and harvested after 72 hr. PP2A trimers were isolated exactly according to *Adams and Wadzinski (2007)*. Protein concentrations were measured in comparison to bovine serum albumin (BSA) standards on silver stained gels. Western blotting confirmed the identity of the protein bands in PP2A preparations. Phosphatase assays with the purified PP2A enzymes were carried out as described above. Enzymes were diluted to the appropriate concentration in PP2A reaction buffer (20 mM Tris, pH 7.5; 1 mg/ml bovine serum albumin; 0.1% β-mercaptoethanol) prior to incubation with the substrate being tested.

Recombinant *Schizosaccharomyces pombe* Fcp1 (wildtype and kinase-dead D172N mutant) proteins were purified after expression in *E. coli* cells as previously described (*Hausmann and Shuman, 2002*; *Kimura et al., 2002*; *Suh et al., 2005*).

## Western blot analysis and antibodies

Samples were prepared by mixing equal volumes of the fraction with 2X SDS loading dye, followed by boiling for 2 min. Samples were then resolved on 10% SDS-polyacrylamide gels. Proteins were transferred to Immobilon-P membranes (Millipore) as previously described (*Castilho et al., 2009*). The following primary antibodies were used to probe Western blots (catalog numbers in parentheses): from Abgent (San Diego, CA): PP6-C (#AP8477b). From Abnova (Taiwan): PP4-C (#PAB14146); and PP5-C (#H00005536-B01). From Cell Signaling Technology (Beverly, MA): PP2A-A (#2309); PP1α (#2582); Twins (PP2A B55 subunit, from clone 100C1; #2290); and FlagTag (DYKDDDDK; #2368). From Santa Cruz Biotechnology: PP2A-Cα, from clone N-25 (#sc-130237); PP1-C, from clone E−9 (#sc-7482); and Fcp1 (from clone H300; #sc32867). From Stratagene (La Jolla, CA): PP2A B′ (B56) pan (#B13009-51). From Thermo Scientific: α-tubulin (#MA5-14992); and striatin (PP2A B‴; #PA1-46460). From Upstate/Millipore (Temecula, CA): PP2A-C subunit, from clone 1D6 (#05-421); and PP2A B subunit, from clone 2G9 (#05-592). Antibodies against the two B56 regulatory subunits of PP2A in *Drosophila* (Widerborst [Wdb] and Well-rounded [Wrd]) were the kind gifts of Dr Timothy Megraw (University of Texas Southwestern Medical Center, Dallas, TX) and have been previously described (*Kotadia et al., 2008*). Antibodies recognizing *Drosophila* Fcp1 (made in rabbits) and TFIIS (made in guinea pigs) were described in *Fuda et al. (2012)*.

The secondary antibodies used were: goat anti-rabbit IgG (H+L)-HRP conjugate (catalog #170-6515; Bio-rad, Hercules, CA) at a 1:5000 dilution; goat anti-mouse IgG (H+L)-HRP conjugate (catalog #170-6516; Bio-rad) at a 1:3000 dilution; and donkey anti-guinea pig IgG (H+L)-HRP conjugate (catalog #706-035-148; Jackson ImmunoResearch Laboratories, West Grove, PA) at a 1:5000 dilution. Chemiluminescent signals were visualized by ECL Western Blotting Substrate (Thermo Scientific) according to the manufacturer's instructions.

## Calculation of $K_m$ and $k_{cat}$ from initial reaction rates at varying substrate concentration

Varying concentrations, $[N]_T$, of $^{32}$P-labeled pEndos (see *Figure 7B,C*) were incubated with $[P]_T = 0.5$ nM PP2A-B55, and the initial velocities, $v$, of $^{32}$P release were measured in (duplicate or quadruplicate) samples in two independent experiments. Technical limitations precluded the use of a PP2A-B55 concentration that was much less than the $K_m$, so the data was analyzed using the Michaelis–Menten reaction scheme with equations that accounted for tight binding. That is, we did not assume that $[N] \approx [N]_T$ or that $[P] \approx [P]_T$. (Unscripted variables are unbound concentrations and the $T$ subscript signifies total concentrations.) Combining the mass conservation and quasi-steady state equations for the concentration of the transient pEndos/PP2A-B55 complex, $[NP]$, gives:

$$[NP] = ([N]_T - [NP])([P]_T - [NP]) / K_m$$

$$v = k_{cat}[NP].$$

This quadratic velocity equation has the solution according to *Morrison (1969)*:

$$v = k_{cat}\left\{([P]_T + [N]_T + K_m) - \sqrt{([P]_T + [N]_T + K_m)^2 - 4[P]_T[N]_T}\right\}$$

The parameters $K_m$ and $k_{cat}$ were determined from the data using nonlinear regression assuming equal fractional errors (Mathematica Version 9), and the weighted means and standard deviations were determined from two independent experiments. The results are displayed in *Table 1*.

## Measurement of $K_m$ by competition of pEndos dephosphorylation with okadaic acid

The measurements described above gave relatively large errors in $K_m$ because of the difficulty of making measurements at the very low PP2A-B55 and pEndos concentrations imposed by the very low $K_m$. To improve accuracy we used a modification of the approach used in *Sasaki et al. (1994)* and *Takai et al. (1995)* to measure the $K_m$ by a supplementary method: competing the PP2A-B55-catalyzed dephosphorylation of a fixed amount of pEndos with varying concentrations of okadaic acid (*Figure 7D,E*). This approach was feasible because the dissociation constant of okadaic acid with respect to PP2A has

been carefully determined (*Takai et al., 1992*; *Sasaki et al., 1994*; *Takai et al., 1995*) and because the affinity of pEndos for PP2A-B55 is in the same order of magnitude. By using an initial pEndos concentration ($[N]_T(0) = 50$ nM or 1 µM, depending on the experiment) that was much greater than the PP2A-B55 concentration ($[P]_T = 0.25$ nM or 1 nM) and by limiting the incubation time, we ensured that only a small fraction of pEndos was dephosphorylated, facilitating a more accurate determination of the $K_m$. Moreover, this ensured that $[P]_T/\{[N]_T(0) + K_m\} < 0.005 << 1$, so the total quasi-steady state approximation that was used or mathematical analysis was good (*Borghans et al., 1996*).

## Mathematical model

Since okadaic acid and pEndos bind competitively to the active site of PP2A-B55, combining the mass conservation equations with the quasi-steady state equations determining the amounts of the pEndos/PP2A-B55 and okadaic acid/PP2A-B55 complexes we get:

$$[N] = \frac{[N]_T}{1 + [P]/K_m}$$

$$[O] = \frac{[O]_T}{1 + [P]/K_d^O}$$

$$[P] = \frac{[P]_T}{1 + [O]/K_d^O + [N]/K_m},$$

where, $[N]$, $[O]$, and $[P]$ are, respectively, the (time-dependent) concentrations of unbound pEndos, okadaic acid, and PP2A-B55, the symbols with $T$ subscripts denote the total concentrations of these compounds, $K_m$ is the pEndos-PP2A-B55 Michaelis constant, and $K_d^O$ ($30 \pm 2$ *nM*) is the okadaic acid-PP2A-B55 dissociation constant. These equations allow for tight binding; that is, we do not assume that $[O] \approx [O]_T$. However, since $[N]_T \gg [P]_T$ at all times, the approximation $[N] \approx [N]_T$ is good. Using this, the equations have the solution:

$$[P] = \left\{ \begin{array}{l} -[O]_T K_m + [P]_T K_m - K_d^O K_m - K_d^O [N]_T \\ + \sqrt{\left([O]_T K_m - [P]_T K_m + K_d^O K_m + K_d^O [N]_T\right)^2 + 4[P]_T K_d^O K_m \left(K_m + [N]_T\right)} \end{array} \right\} \Bigg/ \left[ 2\left(K_m + [N]_T\right) \right].$$

The rate of pEndos dephosphorylation is determined by the total quasi-steady state equation from *Borghans et al. (1996)*:

$$\frac{d[N]_T}{dt} = -k_{cat} \frac{[N][P]}{K_m} \approx -k_{cat} \frac{[N]_T [P]}{K_m}.$$

This equation was numerically integrated using Mathematica Version 9, keeping in mind that $[N]_T$ and $[P]$ are time-dependent, to determine the total amount of dephosphorylation.

Experiments were performed to measure the total amount of dephosphorylation at varying values of $[O]_T$ and fixed values of $[N]_T$, $[E]_T$, and $t$, and weighted nonlinear regression was used to estimate $K_m$ and $k_{cat}$. The weighting factors were set assuming that the fractional errors in the experimental measurements were identically distributed. The results are displayed in *Table 1*.

## Time-dependent inhibition by pEndos of PP2A-B55 dephosphorylation of pCDKS

*Figure 8B*: parallel aliquots containing $[P]_T(0) = 0.25$ nM PP2A-B55, $[pC]_T(0) = 0.47$ µM 32P-phosphorylated pCDKS, and either no pEndos, $[N]_T(0) = 16$ nM non-radioactive pEndos, or 16 nM thio-pEndos were incubated for the indicated times and the amount of 32P released was determined by scintillation counting after removing the pCDKS by TCA precipitation. Because the pCDKS concentration was much less than its $K_m$, 71–99 µM (*Table 1*), it bound a negligible fraction of PP2A-B55; therefore, the analysis would have been identical to that of *Figure 10A* except that PP2A-B55 activity decreased continuously, even in the no pEndos control, presumably due to

degradation or instability. This decrease was modeled assuming a constant degradation rate, $k_{deg}$, so $d[P]_T/dt = -k_{deg}[P]_T$ and $[P]_T(t) = e^{-k_{deg}t}[P]_T(0)$. The fraction of pCDKS dephosphorylated was small, so we approximated $[pC]_T(t) = [pC]_T(0) \equiv [pC]_T$. Therefore, the amount of $^{32}P$ released from pCDKS in the no-pEndos control was:

$$\frac{d\left[^{32}P\right](t)}{dt} = \left(k_{cat}^{pCDKS}/K_m^{pCDKS}\right)[pC]_T[P]_T(t)$$

$$\left[^{32}P(t)\right] = \left(k_{cat}^{pCDKS}/K_m^{pCDKS}\right)[pC]_T[P]_T(0)\left(1-e^{-k_{deg}t}\right)/k_{deg},$$

where, the second line was computed by integrating the first line. An excellent fit was obtained using the values determined by nonlinear regression: $\frac{k_{cat}^{pCDKS}}{K_m^{pCDKS}} = 0.52 \pm 0.01/(\mu M\ sec)$ and $k_{deg} = 0.020 \pm 0.001/min$.

The analysis in the presence of pEndos used the quasi-steady-state equations used for *Figure 10A* except that they were modified by the replacement $[P]_T \rightarrow [P]_T(t)$ to account for the instability of PP2A-B55. $[P]_T(t)$ was computed using the value of $k_{deg}$ as described except that the onset of degradation was postponed for 25 min to accord with the data, which shows that the PP2A-B55 activity (determined by the slope of the curve) after release from pEndos inhibition (e.g., at t ~ 50 min) is greater than that in the no-pEndos control. (This might have been due to stabilization of PP2A-B55 by pEndos binding but, in any case, does not significantly affect the analysis.) These equations along with the equation above determining $\frac{d[^{32}P]}{dt}$ and the value of $\frac{k_{cat}^{pCDKS}}{K_m^{pCDKS}}$ determined in the no-pEndos experiment were numerically integrated to determine $\left[^{32}P\right](t)$ as a function of $K_m$ and $k_{cat}$ These values, $K_m = 0.47 \pm 0.14$ nM and $k_{cat} = 0.03 \pm 0.0005$/sec, were determined by nonlinear regression to the data.

## Theoretical time-course of dephosphorylation of pEndos and activation of PP2A/B55 at the end of M-phase

*Figure 10A*: the values for pEndos PP2A-B55 binding of $k_{on} = 0.057$/(sec nM) and $k_{off} = 0.0068$/sec were determined from the experimentally measured $K_d$ (for thiophosphorylated Endos) and $k_{cat}$. Since $\varepsilon = k_{cat}[P]_T/\left\{k_{on}([N]_T(0)+[P]_T+K_m)^2\right\} = 5.6\times10^{-7} \ll 1$, the total quasi-steady state approximation is expected to be good except for a transient during the small time interval $0 \le t \le t_c$, where $1/\{(k_{on}([N]_T(0)+K_m)\} \approx 0.02$ s (*Borghans et al., 1996*). Therefore, we used the total quasi-steady state equations:

$$[N](t) = \frac{[N]_T(t)}{1+[P](t)/K_m}$$

$$[P](t) = \frac{[P]_T}{1+[N](t)/K_m}$$

$$\frac{d[N]_T(t)}{dt} = -k_{cat}[P](t)[N](t)/K_m.$$

These were numerically integrated using Mathematica Version 9 with $[P]_T = 250$ nM, $k_{cat}^X = 0.05$/sec, and $K_m^P = 1.0$ nM with the boundary condition $[N]_T(0) = 1\ \mu M$.

To be certain that the quasi-steady state assumption was valid over the entire time course (i.e., including the period when $[N](t) < [P](t)$), we recomputed the plots using the explicit mass-action equations for $d[N]/dt$, $d[P]/dt$, and $d[NP]/dt$ (where NP denotes the PP2A-B55/pEndos transient complex). These plots were visually indistinguishable from those computed using the quasi-steady state assumption except, as expected, for a brief transient during $0 \le t \le t_c$.

*Figure 10B*: We assume that the kinetic constants for the additional phosphatase PPTX are $K_m^X = 85$ $\mu M$ and $k_{cat}^X = 22.5$ s$^{-1}$, which are based on the values for PP2A dephosphorylation of CDKS (*Table 1*). Since $K_m^X$ is so large, it is certain that $\varepsilon \ll 1$ (see definition above) so the total quasi-steady state assumption can again be used. In this case, the equations are:

$$[N](t) = \frac{[N]_T(t)}{1 + [P](t)/K_m^P + [X](t)/K_m^X}$$

$$[P](t) = \frac{[P]_T}{1 + [N](t)/K_m^P}$$

$$[X](t) = \frac{[X]_T}{1 + [N](t)/K_m^X}$$

$$\frac{d[N]_T(t)}{dt} = -\left\{ k_{cat}^P [P](t)/K_m^P + k_{cat}^X [X](t)/K_m^X \right\}[N](t),$$

where, $[X]$ and $[X]_T$ are the concentrations of unbound and bound phosphatase X, and superscripts $P$ and $X$ on the kinetic parameters $k_{cat}$ and $K_m$ identify the relevant enzymes. These equations were numerically integrated.

*Figure 10C and D*: When PP2A/B55 is assumed to bind, but not dephosphorylate, pEndos with dissociation constant $K_d^P$, the equations are the same with the substitutions $K_m^P \rightarrow K_d^P$ and $k_{cat}^P \rightarrow 0$. These equations were numerically integrated using $K_d^P = 0.12$ nM, which was based on the dissociation constant of the pEndos thiosulfate phosphomimetic (computed from the data shown in *Figure 8*).

## Fraction of anti-Endos activity coming from PP2A/B55 and PPX

*Figure 10—figure supplement 1*: the first three equations described above for *Figure 10B* were solved to determine $[N]$, $[P]$, and $[X]$ as functions of $[N]_0$ (using the notation above). The amounts of dephosphorylating velocity coming from PP2A/B55 and PPX were then:

$$v^P = k_{cat}^P [P][N]/K_m^P$$

$$v^X = k_{cat}^X [X][N]/K_m^X$$

and the fractional velocities were:

$$f^P = \frac{v^P}{v^P + v^X}$$

$$f^X = \frac{v^X}{v^P + v^X}.$$

## Acknowledgements

We thank the individuals listed in the 'Materials and methods' for providing antibody reagents and genetic stocks.

## Additional information

### Funding

| Funder | Grant reference number | Author |
|---|---|---|
| National Institutes of Health | GM048430 | Michael L Goldberg |
| National Institutes of Health | GM051366 | Brian E Wadzinski |
| National Institutes of Health | DK070787 | Brian E Wadzinski |

The funders had no role in study design, data collection and interpretation, or the decision to submit the work for publication.

## Author contributions

BCW, JJF, KAB-H, DS, Conception and design, Acquisition of data, Analysis and interpretation of data, Drafting or revising the article; BEW, Conception and design, Analysis and interpretation of data, Drafting or revising the article, Contributed unpublished essential data or reagents; NJF, Acquisition of data, Drafting or revising the article, Contributed unpublished essential data or reagents; MLG, Conception and design, Analysis and interpretation of data, Drafting or revising the article

## Ethics

Animal experimentation: This study was performed in strict accordance with the recommendations in the Guide for the Care and Use of Laboratory Animals of the National Institutes of Health. All of the animals were handled according to approved institutional animal care and use committee (IACUC) protocols. Frogs are maintained according to practices prescribed by the NIH and monitored by the Cornell Institutional Animal Care and Use Committee (IACUC) according to protocol 2006-0097, last approved 07/11/2012. The animal quarters are maintained by the Center for Research Animal Resources (CRAR) at the Cornell College of Veterinary Medicine, under the supervision of trained staff and a veterinary doctor.

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
