## [Decision Letter]

Thank you for sending your work entitled “Greatwall-phosphorylated Endosulfine is both an inhibitor and a substrate of PP2A-B55 heterotrimers” for consideration at *eLife*. Your article has been favorably evaluated by a Senior editor and 4 reviewers, one of whom is a member of our Board of Reviewing Editors.

The Reviewing editor and other reviewers discussed their comments before we reached this decision, and the Reviewing editor has assembled the following comments to help you prepare a revised submission.

In this study the authors have addressed the important question of how great wall kinase-dependent inhibition of PP2A phosphatases is reversed upon mitotic exit. The authors use a number of systems (*Xenopus* extracts, *Drosophila* and mammalian tissue culture cells) to investigate the dephosphorylation of Endos, the protein that becomes an inhibitor of the PP2A-B55 isoform of PP2A once Endos is phosphorylated by Greatwall. The authors use a variety of inhibitors and siRNA treatments to identify the phosphatase as PP2A-B55 itself. They resolve this apparent paradox by suggesting that the kinetics of Endos dephosphorylation mean that it will be preferentially dephosporylated when Greatwall is inactivated in mitosis. This study presents an interesting idea for how mitotic exit can be controlled that should be of general interest in the field, and to those working on phosphatases in general. Before publication, however, the following points should be addressed.

1) The description of the phosphatase assays is unclear and should be described in more detail. It would be useful to state that, for instance, whether the dephosphorylation of substrates shown in Figure 2 is linear with respect to enzyme concentrations. In some cases (i.e., data for fostriecin in Figure 2 compared to Figure 6, incorrectly described as Figure 6 in the figure legend) there are different sensitivities to inhibitors that are ascribed to loss of potency with storage – obviously it would be better to repeat using new inhibitor if possible, even thought the conclusions of the experiment are not affected qualitatively. Moreover, it is not clear that Michaelis-Menton kinetics are really appropriate to analyse the substrate-enzyme relationship of a strongly-bound component of an inhibited complex. Can the authors comment? Presumably the dephosphorylation of endosulfine in the complex with PP2A-B55 would result in the activation of the enzyme towards other substrates after incubation. Does this occur? Indeed, dephosphorylation of endosulfine within the complex would presumably mean that the rate would be independent of the dilution of the complex. Is there any evidence that this is the case?

2) Most of the data are presented as single points on a graph, therefore we have no means to judge the variation in the assays. The authors should specify the number of independent experiments for each figure and estimate the variation between experiments.

3) Although the model of “inhibition by unfair competition” looks very beautiful, particularly in implying that the system has an intrinsic reset mechanism, it is assumed that at exit from M-phase, Gwl is inactivated prior to the reactivation of PP2A-B55. Nonetheless, presumably the inactivation of Gwl requires active/reactivated PP2A-B55, because Gwl is activated through direct phosphorylation by CDK1 (as previously reported by the authors). The authors should discuss why and how Gwl inactivation at anaphase onset is anticipated to occur prior to, or in the absence of, reactivation of PP2A-B55.

4) For their proposed mechanism to be valid, pEndos has to present in molar excess over active PP2A-B55 in mitosis because otherwise there would be free PP2A-B55 available to dephosphorylate pCDKS. The authors should measure the amounts of phospho-Endos to PP2A-B55 to address this point.

5) The authors should discuss the caveat that they are using asynchronous mammalian cells to measure an activity that conceivably is most active only for a short time in the cell cycle, i.e. in anaphase. Thus, it is conceivable that they may not have detected the physiological phosphatase in their experiments. They should also comment upon whether the mechanism that they propose is rapid enough to account for the reactivation of phosphatases in mitotic exit.

---

## [Author Response]

*1) The description of the phosphatase assays is unclear and should be described in more detail. It would be useful to state that, for instance, whether the dephosphorylation of substrates shown in*
Figure 2
*is linear with respect to enzyme concentrations*.

We have included a new figure supplement (Figure 6—figure supplement 2) that characterizes the assays we have performed with purified enzymes in terms of linearity with both enzyme amount and time of incubation. For both the pEndos and pCDKS substrates, we took care to ensure that all of the experiments using these enzymes were done in the linear range with respect to enzyme concentrations (less than 1 nM), to time of incubation (less than 15 min), and to substrate utilization (less than 25% of the input phosphate was released).

The reviewers asked specifically whether the assays shown in Figure 2 are linear with respect to enzyme concentrations. In contrast with the experiments shown in Figure 6 and later, which were done with purified enzymes, those in Figure 2 were performed in concentrated extracts. Because of the well-known dilution effects shown in Figure 2 and the new Figure 2—figure supplement 5 (particularly for the pCDKS substrate), we cannot change the concentrations of the enzymes within the extracts without also changing the concentration of endogenous inhibitors (such as pEndos) at the same time; linearity with respect to the extract concentration therefore cannot be expected. However, we do know that these assays were conducted in the linear range with respect to enzyme amounts (the greater the volume and thus the more enzyme, the greater the signal), time, and substrate utilization. We added a statement to this effect in the Figure 2 legend, but we don’t believe another supplemental figure is required given the documentation in Figure 6—figure supplement 2 for cases with the purified enzymes.

*In some cases (i.e., data for fostriecin in*
Figure 2
*compared to*
Figure 6*, incorrectly described as*
Figure 6
*in the figure legend) there are different sensitivities to inhibitors that are ascribed to loss of potency with storage – obviously it would be better to repeat using new inhibitor if possible, even thought the conclusions of the experiment are not affected qualitatively*.

The reviewers were referring specifically to the inhibitor fostriecin, which is somewhat notorious in being unstable (see the reference in the manuscript to the Swingle et al. review). We obtained new fostriecin from two different suppliers, and found that the potency of the material we received was lower than advertised. Both of these samples were encased in exactly the same type of vials; we suspect that both suppliers were selling the same preparation from a single manufacturer that was made many years ago. We also resuspended the fostriecin powder in different diluents (water, DMSO, and ethanol) and found identical results. It should be noted that the potency was reduced for the reactions of both pEndos and pCDKS substrates with the purified PP2A-B55 holoenzyme (Figure 6). We also checked fostriecin against commercially obtained PP2A dimeric enzyme reacted with myelin basic protein as the substrate, and again the potency was much reduced relative to our expectations.

As pointed out by the reviewers, the critical point from the perspective of this paper is that it is the qualitative patterns that matter to our conclusions, not the absolute value. Our experiments all contained internal controls that allowed us to compare the activity of PP2A-like enzymes (sensitive to fostriecin) with that of PP1-like enzymes (relatively insensitive). In this regard, we also checked both of our new fostriecin samples on various preparations of PP1, which were all much more resistant to the inhibitor than were our various PP2A enzymes; this is the expected result.

We have noted these issues briefly in the Materials and methods and also the legend to Figure 6, but we think there is little point in presenting all of the data with respect to all the samples of inhibitors and enzymes when the results already shown are representative.

*Moreover, it is not clear that Michaelis-Menton kinetics are really appropriate to analyse the substrate-enzyme relationship of a strongly-bound component of an inhibited complex. Can the authors comment*?

In the original manuscript, some of the kinetic parameter measurements for pEndos dephosphorylation(those done as a competition with okadaic acid; Figure 7) were actually calculated with a modified form of the Michaelis–Menten equation that accounts for tight binding as the reviewer suggests, but other assays (Figure 7) were done with commercial software that probably used normal Michaelis-Menten assumptions (the documentation is unclear). As described in the revised text, we have recalculated the latter accounting for the tight binding and present these revised calculations in Table 1. In fact, this change brings the two sets of values in somewhat closer correspondence and slightly tightens our estimate for the range of possible values of the pEndos dephosphorylation *K*_*m*_.

In this regard, we were recently amazed to find by searching the BRENDA enzyme database that the *K*_*m*_ for this reaction may be the lowest recorded for any enzymatic process. We have added a sentence to this effect in the manuscript, as it may help readers comprehend the difficulties in these measurements and the unique nature of the pEndos dephosphorylation mechanism.

*Presumably the dephosphorylation of endosulfine in the complex with PP2A-B55 would result in the activation of the enzyme towards other substrates after incubation. Does this occur*?

The reviewers are absolutely correct that after pEndos has been dephosphorylated to a level below that of the concentration PP2A-B55, the enzyme should direct its attention to other substrates; this is in fact the key point to our model. Figure 7 of the original manuscript (now Figure 8) presented some experimental evidence in support of this assertion, but the demonstration of this crucial point was not as direct as it could be. We thus took the suggestion of the reviewer to heart and conducted a time course in which PP2A-B55 was mixed with both radioactive pCDKS substrate and non-radioactive pEndos. These results are shown in new Figure 8 of the revised manuscript, and they correspond extremely closely to our expectations. The enzyme barely works on the pCDKS substrate at early time points. But later, when the kinetics predicts the pEndos would have been substantially dephosphorylated, the rate of pCDKS dephosphorylation increases dramatically to about the same initial rate as the controls. In our opinion, this is a very clear demonstration of the PP2A-B55 reset mechanism we proposed. We particularly thank the reviewers for this suggestion.

*Indeed, dephosphorylation of endosulfine within the complex would presumably mean that the rate would be independent of the dilution of the complex. Is there any evidence that this is the case*?

We agree with the reviewers that such would be the expectations of the system, but it is technically extremely difficult to demonstrate this point to our satisfaction. We did perform one new experiment in which we mixed together the enzyme and pEndos substrate at high concentrations, and then we diluted the sample through a 25-fold range. As expected, the velocity of the reaction was unchanged, meaning that the same number of cpm of phosphate were released regardless of the final concentrations. The problem with this experiment is that it was done at a considerable excess of pEndos to the enzyme, and at the final dilution, the concentration of pEndos was roughly 20 nM. This value is considerably above the *K*_*m*_ and in the range where the enzyme remains saturated with substrate. Basically, this result is just another way of saying that the system is already at Vmax at pEndos concentrations higher than about 10 nM; this conclusion was already presented in Figure 7 in a slightly different way. We thus have not included a figure showing the new less-than-ideal experiment, though we could of course do so if desired.

The experiment we think the reviewers were actually suggesting is probably one in which the enzyme is in considerable excess of the substrate. For example, we could increase the concentration of enzyme to say 5 nM (10 times what we normally use) and decrease the concentration of pEndos to 1 nM. But the sample would then have very few total cpm, and the complex would then have to be significantly diluted which presents difficulties in sample handling. (With higher substrate concentrations, we could avoid the sample handling problem by assaying the same volume of each dilution, but the same approach with more dilute substrate would yield samples with no counts.) An additional problem is that if the enzyme is in excess, the reaction would exhaust almost all of the substrate very rapidly (less than a half minute) even though the *k*_*cat*_ is relatively low. We thus believe that the data that could be gained from such an experiment would not be very believable.

*2) Most of the data are presented as single points on a graph, therefore we have no means to judge the variation in the assays. The authors should specify the number of independent experiments for each figure and estimate the variation between experiments*.

The reviewers are referring to Figure 2 and its various supplements and to Figure 6, which show single points; in contrast, Figures 3, 4, 6 and 7, and the three supplements to Table 1 in the original manuscript did provide means to judge variations. In the case of Figures 2 and 6, we had conducted many additional experiments that convinced us the conclusions were quite reproducible, and the experiments themselves have internal controls for reproducibility in terms of closely spaced titrations. In some ways, we consider the titration points to be more informative than exact replicates of a given assay. Titrations are better able to show patterns that change with concentrations, and the consistency we see between closely spaced points provides a good indication of reproducibility. Nonetheless, we have taken a variety of additional measures to address this issue.

A brief digression: there are several types of experimental replications that may be considered. The first of these concern what we term *technical replicates* (independent assays of the same reagents). In our hands, these technical replicates are almost always within 10-15% of the average value. Next are what we term *biological replicates* (assays using independent preparations of one or more reagents). Finally, we define *evolutionary replicates* as assays using samples (in our case, cell extracts) obtained from different species.

In the revised manuscript, we have responded to the reviewers’ well-taken point in the following ways. (1) We define these types of replicates in the Materials and Methods, and then give the type and number of replicates for each individual experiment in the legend for the figure describing that experiment. (2) For Figure 2 and its supplemental figures, we have included several new figures that show biological and evolutionary replicates that reveal very similar patterns, demonstrating their reproducibility. The new figures added to the revised manuscript are:Figure 2—figure supplement 1 parts A B EFigure 2—figure supplement 2 parts E and FFigure 2—figure supplement 3 parts B and DFigure 2—figure supplement 5 (this figure supplement is completely new).

In addition, we altered a few of the figures by adding error bars or additional entries representing the technical replicates that were previously performed but had not been shown:Figure 2—figure supplement 1 part DFigure 2—figure supplement 2 part C.

(3) For Figure 6, we added a new part E that shows technical replicates and error bars for the reactions performed either without inhibitors or at the highest concentration of each of the inhibitors.

*3) Although the model of “inhibition by unfair competition” looks very beautiful, particularly in implying that the system has an intrinsic reset mechanism, it is assumed that at exit from M-phase, Gwl is inactivated prior to the reactivation of PP2A-B55. Nonetheless, presumably the inactivation of Gwl requires active/reactivated PP2A-B55, because Gwl is activated through direct phosphorylation by CDK1 (as previously reported by the authors). The authors should discuss why and how Gwl inactivation at anaphase onset is anticipated to occur prior to, or in the absence of, reactivation of PP2A-B55*.

This is indeed an important issue and one that we cannot completely answer at this point in time because we do not know what phosphatases are responsible for Gwl inactivation. At the outset of our experiments, we had the same supposition as the reviewers: that PP2A-B55 would be the main such enzyme because CDK phosphorylations activate Gwl and PP2A-B55 can remove certain such phosphorylations. However, it is increasingly clear that PP2A-B55 does not act on all, or even perhaps the majority, of CDK sites. The best evidence for this statement comes from a recent paper from Cundell et al. that was referenced earlier by the reviewers. Thus, there is no a priori reason to conclude that Gwl must be inactivated by PP2A-B55.

Our own studies point to the existence of at least one Gwl-inactivating phosphatase that is not PP2A-B55. (i) Purified PP2A-B55 that we know is active against pCDKS and pEndos substrates is very inefficient at inactivating Gwl in vitro; in fact, after 90 minutes of incubation with a molar amount of phosphatase in excess to that of Gwl, more than 70% of Gwl activity remained. (ii) When we add active Gwl to extracts (frog eggs, or HeLa or S2 cells) that are not in M phase, Gwl rapidly loses its phosphorylations and is turned off even in the presence of high concentrations of okadaic acid that would completely inhibit PP2A. We presume that this okadaic acid-insensitive phosphatase is driving Gwl inactivation during M phase exit, but we do not yet know the identity of this enzyme. This phosphatase could be constitutively active if the rate of Gwl phosphorylation by CDK1 during M phase is higher than the rate of Gwl dephosphorylation, or this enzyme could conceivably be negatively regulated directly or indirectly by CDK1.

We have added a paragraph to the Discussion that addresses this key point and presents a short version of the arguments just made.

The recently-published paper by Hegarat et. al. claims to have identified PP2A-B55 as the major Gwl-inactivating enzyme; however, they also state that an okadaic acid-resistant phosphatase contributes to Gwl inactivation. We have serious concerns about their data on this point and their results concerning the identity of the anti-Endos phosphatase. For example, the activity they measure in PP2A-B55 IPs that can remove unspecified phosphates from Gwl (which may or may not be involved in Gwl regulation) is less than 30% higher than a high background (their Figure 5). We have included a discussion of this recent publication and how the results are inconsistent with our own.

*4) For their proposed mechanism to be valid, pEndos has to present in molar excess over active PP2A-B55 in mitosis because otherwise there would be free PP2A-B55 available to dephosphorylate pCDKS. The authors should measure the amounts of phospho-Endos to PP2A-B55 to address this point*.

We have not ourselves measured the amount of phospho-Endos, but we have good reason to assert that its concentration during M phase is indeed higher than that of PP2A-B55.

(i) This must be the case because the rate of dephosphorylation of several CDK substrates is decreased almost to 0 during M phase, as we and S. Mochida and T. Hunt have shown in several publications (a repeat of these experiments is also shown in Figure 2 of this manuscript).

(ii) We have measured the relative concentrations of total PP2A-B55 (Twins) and Endos in fly S2 cells, and the relative stoichiometry is 1:5. We document these results in new Figure 2—figure supplement 6. Cundell et al*.* have arrived at a very similar figure, and we cite this reference. In addition, we summarize in the revised text information culled from the Pax-DB database of protein abundance levels determined by global mass spectrometry. These data show that Endosulfine proteins are present in more than a 2-fold molar excess to PP2A-B55 in several different organisms.

(iii) What fraction of this excess of Endosulfine proteins is phosphorylated at the Gwl site during M phase? We could not easily answer this question because we do not have antibodies specific to the phosphorylated form of Endos that are sufficiently sensitive to detect endogenous levels. However, S. Mochida and collaborators have previously shown by a combination of Western blotting and phos-tag gels that in frog M phase extracts, Endosulfine proteins are nearly quantitatively phosphorylated. Using similar methods, but on extracts from synchronized tissue culture cells, the Cundell et al. paper showed that roughly 50% of the Endosulfine was phosphorylated. Even with the lower of these estimates (which may reflect the presence of some cells that were not at M phase at the time), sufficient headroom exists to conclude that the molar concentration of pEndos during M phase in fact exceeds that of the PP2A-B55 phosphatase.

We summarize these arguments in the Discussion of the revised manuscript.

*5) The authors should discuss the caveat that they are using asynchronous mammalian cells to measure an activity that conceivably is most active only for a short time in the cell cycle, i.e. in anaphase. Thus, it is conceivable that they may not have detected the physiological phosphatase in their experiments. They should also comment upon whether the mechanism that they propose is rapid enough to account for the reactivation of phosphatases in mitotic exit*.

We are somewhat puzzled by the reviewers’ comments here. An entire section of the Discussion is titled “Rapid M phase exit requires the inhibition by unfair competition mechanism”. In this section, we present in some detail arguments showing first that our proposed mechanism allows for the reactivation of PP2A-B55 within a couple of minutes (or even less than a minute if other phosphatases participate in reducing the concentration of pEndos to that of the phosphatase), and second that any other mechanism of pEndos inactivation would require far too much time for mitotic exit to be accomplished. We don’t see how we can add further to our presentation in this regard.

We have added to the Discussion a paragraph considering the caveat desired by the reviewer. However, we do not actually believe that the anti-Endos phosphatase is active only during the short time of anaphase. It is our contention that PP2A-B55 acts constitutively throughout the cell cycle against pEndos (although pEndos restricts PP2A-B55 activity against other substrates during M phase). Note in Figure 2 that the anti-Endos activity is present during M phase and appears to increase about 2-fold during interphase. This 2-fold difference is likely artifactual, being due to the presence (M phase) or absence (interphase) of unlabeled pEndos in the extract, which could compete with the exogenous labeled pEndos substrate. Moreover, the characteristics of the M phase and interphase anti-Endos activities are very similar (Figure 2—figure supplement 1; Figure 2—figure supplement 2; and data not shown), and thus almost certainly represent the same enzyme (namely PP2A-B55).

We see no reason to posit the existence of an additional enzyme active only during the transition between M phase and interphase if PP2A-B55 is active against pEndos under both conditions and if the level of this activity, as we have shown, is by itself sufficient to account for the speed of M phase exit.